# Research

ecology, behaviour, environmental science

foraging cues, seabirds, oceanographic features, turbulence, unmanned aerial vehicles, hidden Markov model

**Author for correspondence:**
Lilian Lieber
e-mail: l.lieber@qub.ac.uk

# A bird's-eye view on turbulence: seabird foraging associations with evolving surface flow features

Lilian Lieber[1], Roland Langrock[2] and W. Alex M. Nimmo-Smith[3]

[1]School of Chemistry and Chemical Engineering, Queen's University Belfast, Marine Laboratory,
12–13 The Strand, Portaferry BT22 1PF, Northern Ireland, UK
[2]Department of Business Administration and Economics, Bielefeld University, Postfach 10 01 31, 33501 Bielefeld, Germany
[3]School of Biological and Marine Sciences, University of Plymouth, Drake Circus, Plymouth PL4 8AA, UK

LL, 0000-0002-4833-9594; RL, 0000-0001-8206-6908; WAMN, 0000-0003-3108-9231

Understanding physical mechanisms underlying seabird foraging is fundamental to predict responses to coastal change. For instance, turbulence in the water arising from natural or anthropogenic structures can affect foraging opportunities in tidal seas. Yet, identifying ecologically important localized turbulence features (e.g. upwellings approximately 10–100 m) is limited by observational scale, and this knowledge gap is magnified in volatile predators. Here, using a drone-based approach, we present the tracking of surface-foraging terns (143 trajectories belonging to three tern species) and dynamic turbulent surface flow features in synchrony. We thereby provide the earliest evidence that localized turbulence features can present physical foraging cues. Incorporating evolving vorticity and upwelling features within a hidden Markov model, we show that terns were more likely to actively forage as the strength of the underlying vorticity feature increased, while conspicuous upwellings ahead of the flight path presented a strong physical cue to stay in transit behaviour. This clearly encapsulates the importance of prevalent turbulence features as localized foraging cues. Our quantitative approach therefore offers the opportunity to unlock knowledge gaps in seabird sensory and foraging ecology on hitherto unobtainable scales. Finally, it lays the foundation to predict responses to coastal change to inform sustainable ocean development.

## 1. Introduction

Understanding how physical processes in our oceans shape the foraging distributions of marine predators is critical to predict responses to environmental change [1,2]. Identifying the drivers of animal foraging movement can also help mitigating the ecological impacts of anthropogenic activities [3]. Coastal environments are undergoing unprecedented anthropogenic change, including the installation of man-made structures supporting the blue economy (e.g. ocean and offshore wind energy extraction, mariculture). This coastal change is undoubtedly leading to new interactions between marine predators and installations. While we are yet to understand how this may influence foraging success [4], there is some evidence that installations can even generate new foraging opportunities [5,6]. Foraging strategies may vary in response to physical changes in local conditions. Assessing how free-ranging animals adjust and fine-tune their foraging movements in highly complex and dynamic environments is therefore fundamental to understand how they may respond to anthropogenic change.

Recent advances in satellite feature extraction and the tracking of coherent oceanographic features (e.g. fronts and eddies) have revealed their influence

on near-surface processes and marine predator associations [7,8]. In the pelagic realm, a diverse range of predators have been shown to associate with sub-mesoscale (less than 10 km) [9] and mesoscale (approx. 10–100 km) physical features [10,11]. Mesoscale eddies can provide physical mechanisms to transport [12] or aggregate prey [13], thereby providing foraging opportunities for wide-ranging marine predators, from seabirds to sharks [14–17].

Apart from mesoscale surface ocean features, it has been shown that important levels of predator aggregations can occur at much finer scales, with short internal waves (approx. 0.1–1 km) playing a major role [18]. In near-coastal regions, even more local (approx. 10–100 m) turbulence features including localized upwellings and eddy vortices can similarly provide profitable foraging opportunities for predators, but have rarely been adequately quantified due to limited observational scales. Such turbulence can provide physical mechanisms to enhance prey accessibility, possibly as a result of prey displacement in the water column, through turbulent vertical transport, or physical aggregation at the surface (e.g. at the edges of features) [6,19–21].

For seabirds, coastal environments provide important foraging opportunities [22,23]. Therein, tidal environments present one of the world's most dynamic and turbulent marine habitats [24]. Here, strong currents interacting with fine-scale heterogeneity in bathymetric features or man-made structures can give rise to numerous physical processes, including localized features (e.g. boils [25] (localized upwellings), convergences, eddy vortices) and dynamic boundary waters (e.g. shear lines and flow reversals [26]). Localized turbulence features such as upwelling boils (regions of positive divergence) or vortices are highly dynamic, evolving and dissipating over time scales of minutes [25]. After erupting at the surface, boils will increase in size, decrease in intensity and may evolve into vortical structures before dissipating. Volant predators, such as seabirds, must therefore be able to locate such physical cues for prey across a highly dynamic range of flow features.

Predator–prey interactions are scale dependent [27], but it remains unanswered how seabirds associate with highly localized (approx. 10–100 m), ephemeral flow features to find prey. Strong hydrodynamic processes ultimately determine the spatial distribution of small prey items and seabirds may show affinity to areas characterized by physical properties that enhance or accumulate resources [28]. Direct measurements of localized predator foraging bouts in relation to dynamic physical features could therefore give novel insight into physical cues underlying foraging strategies. Yet, the required high spatio-temporal resolution (metres and seconds) to capture such associations is often unattainable using traditional approaches, such as associating coarse-scale satellite-derived data with higher resolution animal telemetry. For instance, with the rapid dynamics associated with seabird flight, temporally or spatially averaged oceanographic data leads to a spatio-temporal mismatch between movement metrics and habitat characteristics at the visited cell and may thus not capture highly localized associations. With emerging technologies to track animals in their natural environment, such as animal-borne GPS tags and accelerometers [29,30], marine radar [31], ornithodolites [32] and unmanned aerial vehicle (UAV) or drone applications [6,33,34], the development of technical innovations that can link high-resolution animal positions

with dynamic, proximate physical cues and variables is at the forefront of understanding where predators forage and why. Specifically, the application of UAV-based approaches can shed new light on individual movement metrics and underlying physical variables, as perceived by the animal. Drone-extracted surface variables and animal displacement could then be adequately projected onto a two-dimensional plane [35]. Such visualizations would allow a 'bird's-eye view' on underlying physical features, thereby aiding the quantification of context-specific behaviours [36].

When deciding where to search intensively or forage for prey, during flight, exclusive surface-foraging seabirds (e.g. gulls and terns) may focus their visual attention either directly below, or towards upcoming coherent features at the water surface which may be indicative of a profitable foraging opportunity. We hypothesized that terns (Sternidae) vary their foraging movement in response to localized coherent surface flow features, predominantly vorticity (the curl of the surface flow) and upwellings (regions of positive divergence/boils), which could serve as physical foraging cues.

Here, foraging terns were tracked across the wake of a monopile structure (similar to wakes of islands experiencing strong tidal flows [37]) by hovering drones. We mapped the terns' trajectories and underlying surface velocity field in synchrony (figure 1). Subsequently, we used these physical covariates within a hidden Markov model (HMM) to quantify tern foraging associations with underlying evolving flow features. Speed and tortuosity of the tracked individuals differentiated two states, active and transit foraging. We predicted that state transition probabilities would be affected by the strength of the underlying turbulent feature as well as its distance, as perceived by the terns. This allowed us to quantify the influence of prevalent oceanographic features on a surface-foraging marine predator on hitherto unobtainable scales (approx. 10–100 m).

## 2. Methods

### (a) Study site

The study was performed in the Narrows, a tidal channel located in between the Irish Sea and Strangford Lough, Northern Ireland, UK. The UAV surveys were performed over the flood-tide wake of a tidal energy structure (SeaGen; 54° 22.144′ N, 5° 32.777′ W), which consisted of a surface-piercing monopile (3 m diameter) fixed on the seabed (water depth approximately 25 m). SeaGen was fully decommissioned on 25 July 2019. At the time of data collection (6 July 2018), it was non-operational and the twin-rotors had already been removed. Characterized by depth-averaged velocity magnitudes exceeding 5 ms$^{-1}$ during spring tides [26], the remaining monopile generated a von Kármán vortex street in the downstream wake, dominated by turbulent flow features including swirling vortices and localized up- and downwelling, similar to turbulence arising from natural wake features. Strangford Lough hosts various nesting colonies of summer-breeding tern species (*Sterna hirundo*, *S. sandvicensis* and *S. paradisaea*), and SeaGen's floodtide wake was identified previously as a foraging hotspot, generating the highest numbers of terns foraging compared to natural wake features investigated in the tidal channel [6].

### (b) Unmanned aerial vehicle surveys

To record fine-scale tern foraging behaviour in relation to underlying coherent flow features, unmanned aerial vehicle (UAV)

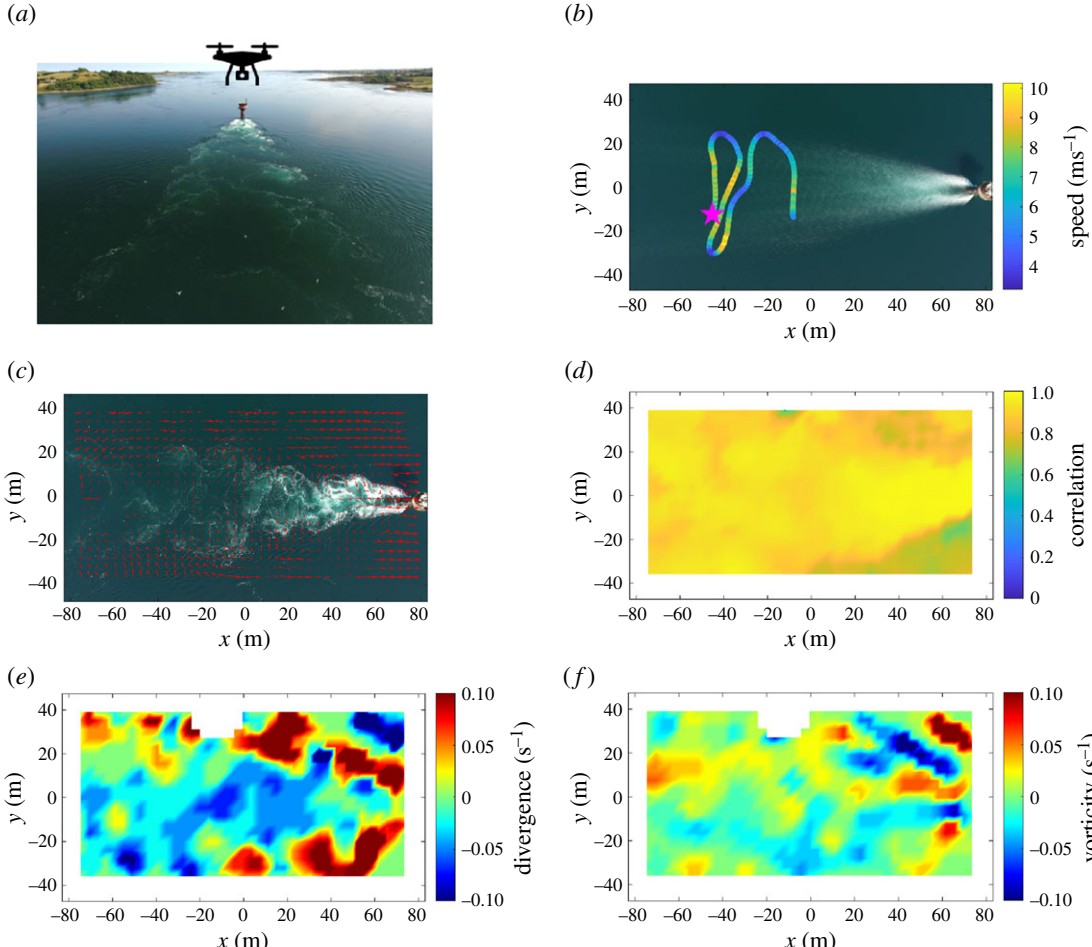

**Figure 1.** Methods overview of data collection, tracking and flow feature extraction. (*a*) Drone hovers were performed at 100 m altitude over a von Kármán vortex street resulting from the floodtide wake of a monopile structure set in a tidal channel. (*b*) Individual terns were tracked using machine learning and manual post-processing (where magenta star marks the start of the track, the wake feature shows a time-average) and flight characteristics (speed and tortuosity) were extracted. (*c*) Surface velocity fields were extracted using PIV techniques. (*d*) A correlation threshold of 0.6 (60%) was used for velocity calculations and subsequent extraction of regions of (*e*) divergence and (*f*) vorticity. (Online version in colour.)

surveys were performed using a DJI Phantom 3 quadcopter recording 2 K video at 30 Hz. The UAV was flown manually using the DJI Go application. In order to comply with best practices and minimize potential disturbance, sampling was performed at a height of $100 \pm 1$ m above-surface level, as measured by the on-board altimeter. Missions included hovers (holding station with a vertically downward-facing camera) varying between 68 and 153 s in duration (total sampling time = 557 s; see electronic supplementary material, table S1) over half a tidal cycle on 6 July 2018 recording the floodtide wake of SeaGen to capture seabird flight tracks over time. All missions were completed in accordance with local regulations and flown by the same qualified (UK Civil Aviation Authority) pilot. The UAV camera was calibrated in the laboratory using a standard checkerboard method and video sequences post-processed using MATLAB (R2017b; Mathworks). At 100 m altitude, each video frame recorded an area of $166.1 \times 94.5$ m$^2$ with the major axis orientated with the mean flow direction.

## (c) Tern tracking, post-processing and extraction of tack parameters

Machine learning approaches were used to identify, count and track terns over the turbulent floodtide wake [6]. Briefly, moving objects were detected using frame-to-frame differencing of the red channel of the raw drone video; red being selected as having the highest contrast to the green water colour. Following

cleaning by dilation and erosion, using a 9-pixel radius disk structuring element, objects were segmented and then filtered by size to remove sun-glint speckles (area < 20 pixels) and large foam patches (area > 500 pixels). Images of potential targets were then passed through a trained 'Bag of Features' classifier [38] before using Kalman filters to compile tracks of those targets identified as terns only. The classifier was trained using 806 manually identified images each of foam and terns, with an average accuracy of 93% when applied to a validation set of 3764 images.

Individual tracks were then subjected to manual quality control. False-positive targets were removed from the track, and tracks were split or truncated where the Kalman filter failed to follow the same target. Track segments were then spliced together, with a subsequent filling of missing targets (electronic supplementary material, figure S1). Following manual post-processing, there were 657 tern tracks in the dataset which were further filtered to only keep those with a minimum of 15 s duration and discarding 'transiting' trajectories. As this study's objective was to analyse terns deemed to be foraging, birds that were solely transiting through the area were identified and excluded. For this, tortuosity, a measure of the curvature of an animal's path (how much the animal is turning), for each overall track was calculated and those with a value less than 1.1 removed (electronic supplementary material, figure S2). Finally, the track positions were corrected for camera lens distortion and scaled according to the UAV's altitude.

The instantaneous velocity and tortuosity along each track were all calculated using an 11 element window (±5 frames, centred on each position), where the raw positions within this window were smoothed by fitting a cubic spline to each window, and the velocity being the first differential of this spline. This represents a low-pass filtering operation with a cut-off frequency at 2.73 Hz. This approach removes higher frequency variation in the instantaneous positions associated with changing body shape during wingbeats (which occur in the frequency range of 3.1 to 3.7 Hz in common and Sandwich terns, respectively [39]). The tortuosity was calculated as the total distance travelled (sum of the distances between the 11 points) divided by straight-line distance between the first and last position.

## (d) Particle image velocimetry

Water surface velocity fields (speed and direction of the flow) were extracted every 0.25 s through each video sequence using particle image velocimetry (PIV) techniques. At each instant, four consecutive video frames were used. The green channel (selected as most representative of the water colour) of each were extracted and then corrected for camera lens distortion. A standard cross-correlation technique, including sub-pixel localization, between consecutive frames was then applied using $65 \times 65$ pixel windows with 50% overlap and 128-pixel clear border [40]. This results in fields of $20 \times 39$ velocity vectors extracted per frame-pair with a correlation coefficient reported for each vector indicating its quality. These raw velocity fields are adversely affected by local spurious artefacts (sun glint, birds). To reduce these effects, a $3 \times 3 \times 3$ median filter was applied across the three vector fields extracted from the four consecutive video frames providing one clean velocity field every 0.25 s through the video sequence that were then scaled according to the UAV's altitude. There were no static reference points within the camera's field of view so that whole-field contamination from the relative motion of the UAV cannot be removed. However, turbulence parameters (vorticity and divergence), derived from local velocity gradients, are minimally impacted by this. These were calculated using the standard MATLAB functions from each velocity field after the application of a minimum correlation threshold of 0.6.

## (e) Matching between tracks and turbulence

Flow parameters (vorticity and divergence) were extracted for each instantaneous position along each track using three-dimensional interpolation in space and time through the corresponding sequence of more sparsely spaced flow fields. A bird's visual perception during foraging is primarily driven by the timing of arrival at a target [41]. However, we did not know a priori if terns would respond to environmental cues directly underneath their flight path or slightly ahead and whether this relationship faded with increased temporal distance to the feature. To investigate such 'time-to-contact' effects, time-offsets (delay $d \in \{0, 0.25, 0.5, \ldots, 5\}$ in seconds) were applied, where $d = 0$ indicates that the vorticity/divergence values were extracted directly underneath the tern's $xy$ position, and $d > 0$ represents values ahead along the tern's flight path. To ensure parity between time-offsets, all tracks were truncated by the maximum offset of $d = 5$.

## (f) Hidden Markov model

We used the extracted physical variables, vorticity magnitude (absolute(curl)) and upwelling (positive divergence), as covariates within a HMM to quantify tern foraging associations with evolving, spatio-temporally explicit surface flow features. When applied to animal movement data, HMMs can reveal underlying (hidden) behavioural states such as 'resting', 'foraging' or 'travelling'

[42,43]. They can further quantify state-switching probabilities as a function of covariates, thereby relating the behavioural states to underlying environmental factors [44,45].

In an HMM [46], a time series of observations is modelled dependent on underlying, non-observable states, with the state sequence evolving according to a Markov chain. We modelled the bivariate time series of tern speed and log(tortuosity) dependent on two underlying states, which could be related to active and transit foraging, respectively (as adapted from definitions of continuous behaviour categories, differentiating between direct flight, active and transit search, applied by JNCC during visual tracking of tern species [47]). Assuming conditional independence of speed and log(tortuosity), given the current state, we used univariate gamma state-dependent distributions for both (non-negative) variables. The evolution of the two states over time, as governed by a two-state Markov chain, was further investigated by relating the state transition probabilities to the covariates absolute(curl) and divergence:

$$\text{logit}(\Pr(i \rightarrow j)) = \beta_0^{(ij)} + \beta_1^{(ij)} \cdot \text{absolute(curl)} + \beta_2^{(ij)} \cdot \text{divergence},$$

for $i,j = 1,2, i \neq j$. To assess how tern state-switching and stationary state probabilities were influenced by the physical variables, also in relation to 'time-to-contact', a range of scenarios were tested within the HMM framework using covariates extracted directly underneath and along the path ($d$). This two-state HMM was fitted in R (R Core Team, 2020) via numerical optimization of the likelihood function, using multiple random initial values as starting points to decrease the risk of missing the global maximum. From the fitted model, stationary state probabilities were extracted using the Markov chain's steady-state distribution under fixed covariate values [43]. The model selection involved AIC comparisons with models without either of the two covariates.

# 3. Results

## (a) Foraging state-dependent distributions

Following the removal of transiting trajectories, there were 143 tern foraging tracks with a minimum of 15 s duration in the dataset, used for subsequent analyses and presented herein. The mean duration of tern tracks was 28.62 s, with a maximum track duration of 98.96 s (see a histogram of track duration in electronic supplementary material, figure S3). Speed and log(tortuosity) of all tracked terns were used as observed variables in the HMM to decompose the tracking data into two states, which could be interpreted as proxies for active (state 1) and transit (state 2) foraging, respectively. Active foraging was indicative of actively searching for food, including instantaneous foraging behaviours of plunge diving and surface feeding, characterized by more erratic flight, including swooping (mean log(tortuosity) = $0.066 \pm 0.068$ s.d.) and lower flight speeds (mean speed = $3.981$ ms$^{-1}$ $\pm$ $1.360$ s.d.). Transit foraging was indicative of opportunistically searching while in transit, characterized with flight speeds faster than active search (mean speed = $7.191$ ms$^{-1}$ $\pm 2.097$ s.d.) and less erratic directional changes (mean log(tortuosity) = $0.009 \pm 0.008$ s.d.). The state-dependent distributions, shown in figure 2a,b, reflect distinct movement patterns for the two behavioural states. Figure 2c,d displays an example track two-dimensional projection and time series, respectively, and associated decoded states.

## (b) Effects of turbulence features on foraging states

The state transition probabilities ($\Pr(i\rightarrow j)$, for $i,j = 1,2$) and as a consequence also the stationary state probabilities ($\Pr(i)$, for $i =$

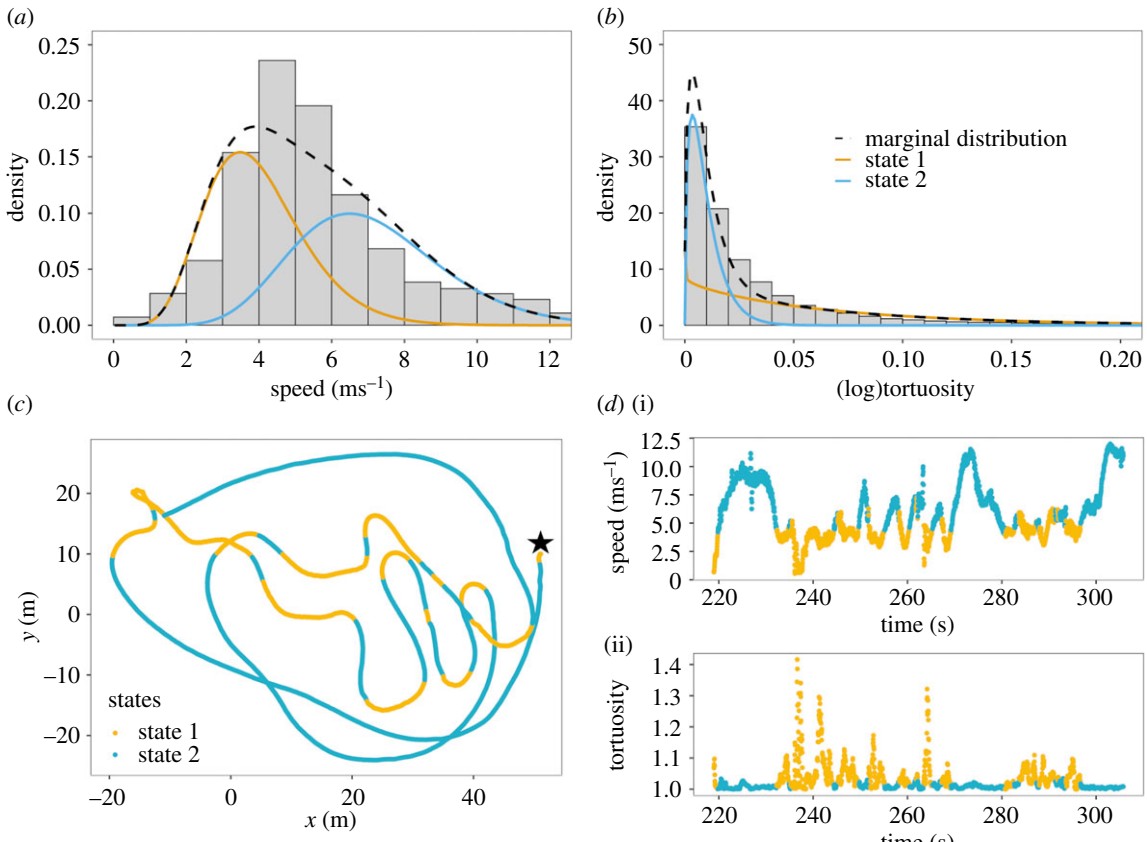

**Figure 2.** HMM fitted to speed and log(tortuosity) data. (*a,b*) Histograms show the observed speed and log(tortuosity) overlaid with the colour-coded state-dependent distributions as estimated for these variables by the HMM (weighted according to proportion of time the corresponding state is active). These were used to identify the two states, active (state 1 = orange) and transit (state 2 = blue) foraging. State 1 is indicative of active search behaviour characterized by lower flight speeds (less than 5 ms$^{-1}$) and more erratic, tortuous movements, including hovers and plunge dives. State 2 is indicative of transit search, characterized by higher flight speeds and less erratic movement, such as during opportunistic searches. (*c*) An example movement track (star symbol demarks the starting location) (*d*) time series along the same track and variation in speed and tortuosity, colour-coded by the predicted behavioural state. (Online version in colour.)

1,2) were modelled as functions of vorticity magnitude (absolute(curl)) and divergence as covariates in the HMM to investigate how active and transit foraging varied with the underlying physical features. The model with both covariates included was favoured by the AIC over models excluding either of the two covariates (ΔAIC = 9.24 for the model without divergence, ΔAIC = 5.46 for model without absolute(curl)).

Following the assessment of the various absolute(curl) and positive divergence (upwelling) delay combinations (time-to-contact), the optimal values (yielding the best fit as measured by the maximum log-likelihood) were $d = 0.25$ for absolute(curl) and $d = 2.0$ for positive divergence (see electronic supplementary material, table S2). It was not known *a priori* how terns would perceive dynamic cues during flight, and these values identified the scales at which the variation in the data, and specifically the probabilistic switching between the two states was best explained. Vorticity extracted almost underneath the terns and divergence ahead of the flight path thus yielded the model with the best goodness-of-fit. This does not necessarily imply that terns primarily respond to features at these time-to-contact values, and several other delay combinations yielded maximum log-likelihood values not much smaller than the optimum. Maximum log-likelihood values were in fact substantially lower when using higher delays $d$ for absolute(curl), but not much lower for any $d$ from 0 to 5 for divergence (electronic supplementary material, table S2).

### (i) State transition probability [Pr(i->j)]

With an increased strength in the vorticity feature underneath ($d = 0.25$), terns were more likely to switch into the active foraging state as depicted in figure 3a [P(2->1)], thus exhibiting shorter travelling bouts (figure 3a; [P(2->2)]). Conversely, for strong positive divergence extracted ahead of the terns' flight paths ($d = 2.0$), the probability of a transition into the active foraging state [Pr(2->1)] decreased (figure 3b); in other words, the sojourn times in the transit foraging state increased with the detection of a distant, strong upwelling (positive divergence) feature (figure 3b; [Pr(2->2)]).

### (ii) Stationary state probability [Pr(i)]

Overall, the probability of terns actively foraging (state 1) increased with the strength of the vorticity features as shown in figure 3c. This relationship was strongest when the vorticity feature was extracted almost directly underneath the tern's position ($d = 0.25$). Conversely, strong positive divergence ahead of the flight path ($d = 2.0$) increased the probability of terns to occupy the transit foraging state (figure 3d).

## 4. Discussion

Our drone-based approach, tracking seabirds and underlying physical features in synchrony, revealed new insights into

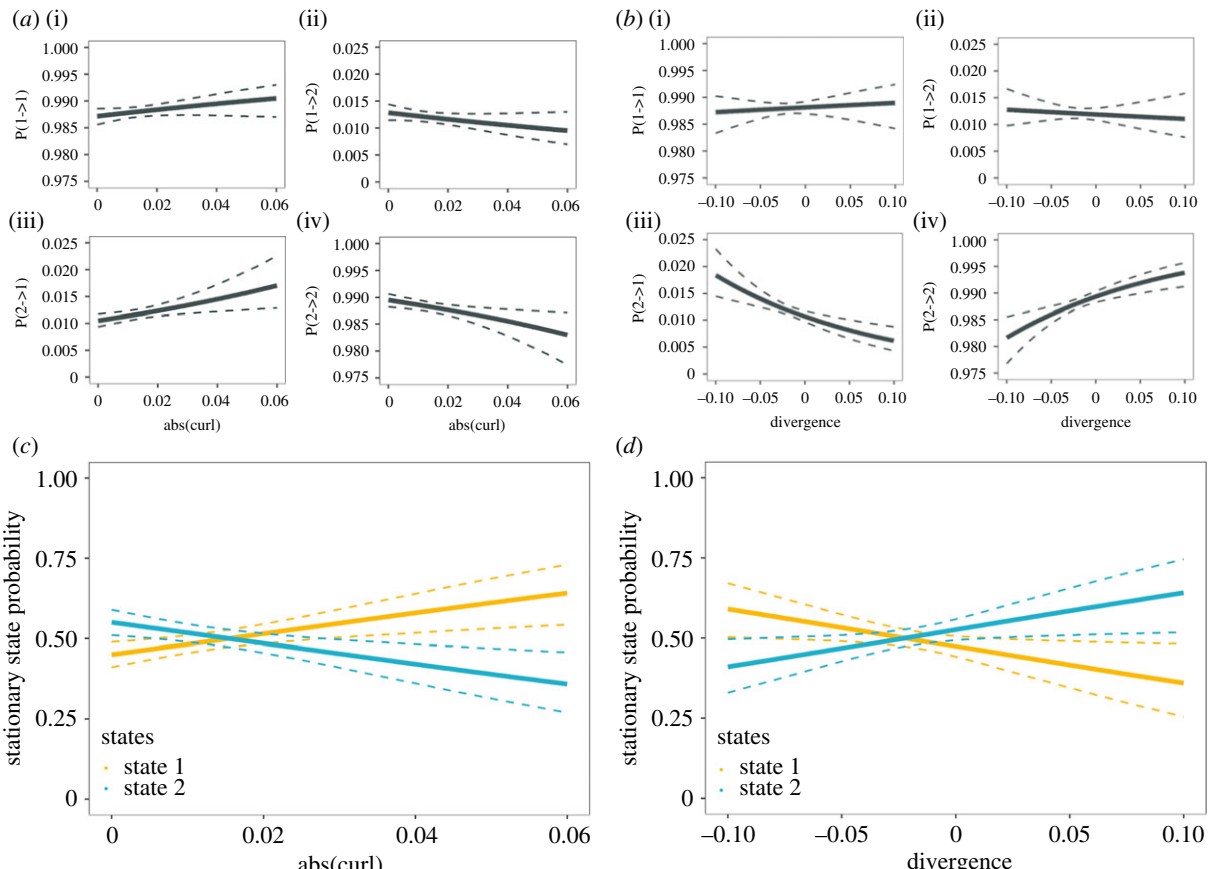

**Figure 3.** State switching and stationary state probabilities as a function of the covariates. (a,b) Probabilities of switching between the two behavioural states, active (state 1, orange) and transit foraging (state 2, blue), respectively, as a function of the covariates absolute(curl) (d = 0.25) and divergence (d = 2.0). (c,d) Stationary state probabilities of occupying the two behavioural states as a function of the covariates. (Online version in colour.)

localized tern foraging strategies among turbulence. We hypothesized that terns may vary their foraging movement in response to localized coherent surface flow features, which could serve as physical foraging cues. As predicted, tern movement patterns showed associations with specific evolving turbulence features, and these varied with the time-to-contact (as expressed in delays), indicating the scale at which most variation in the data was explained.

Terns were more likely to switch to (and occupy) the active foraging state as near-underlying (d = 0.25) vorticity magnitude increased (figure 3a,c). Regions of strong vorticity, patches of swirling flow, tend to accumulate buoyant material at their centres due to secondary circulation patterns [48], which could explain the importance of underlying vorticity to cue active foraging behaviour. Further, terns tended to increase the occupancy of the transit foraging state when strong positive divergence (upwellings/boils) laid ahead of their movement paths (d = 2.0). Therefore, conspicuous upwellings may provide a strong physical cue even at some distance, leading to the investigation of such features. This makes sense in terms of visual perception, as newly erupting boils (strong positive divergence) are easily detectable from a distance, producing smooth patches at the sea surface. However, these flow structures are continuously evolving. For instance, a few seconds after a boil erupts, it will increase in size at the water surface, with surface convergences, associated downwelling and vortex structures evolving at its peripheries [25]. Over time, buoyant material (e.g. small prey items) will accumulate over and within any region of local downwelling. This means that the same boil, on

approach, will already have changed in scale, intensity and distribution of potential prey items. Conspicuous boils have previously been hypothesized to be linked to foraging activity [24], but until now, we have lacked the high spatio-temporal resolution to quantify this adequately. Data gaps remain, specifically, the ecological importance of upwelling boils and how they may contribute to foraging success. Therefore, future research will investigate if seabirds preferentially forage within boils [19], target the edges of boils, or if seabirds may even track boils during flight, taking into account various wind and flow conditions (e.g. electronic supplementary material, figure S4).

While several mechanisms may be in place for seabirds to switch into an active foraging state (e.g. intra- and inter-specific kleptoparasitism 'prey stealing' [49], conspecific attraction or 'local enhancement' [50,51]), our findings indicate that terns are likely to adjust their foraging strategies to localized physical cues at the sea surface. In fast-moving tidal flows, surface-foraging predators must locate patchily distributed prey that moves in space and time, thereby constantly adjusting their behavioural strategies in relation to underlying physical cues.

Previous studies have found seabirds tracking more persistent oceanographic features on larger spatio-temporal scales [9], however, the underlying mechanisms in this study's findings may differ substantially. For instance, it has been found that procellariform seabirds may use olfaction-mediated foraging to track high concentrations of dimethyl sulfide, where olfactory landscapes mark large-scale areas where prey is likely to be found [52]. However,

these odour cues are suggested to operate at larger scales and have not been investigated in tern species. At the localized scales we investigated, terns are more likely to rely on visual cues rather than on olfaction (biogenic cues), alone. Our observed movement associations with underlying vorticity and at a distance, positive divergence, might offer some insight into the visual sensory ecology of terns, as localized and ephemeral by nature, these features could be regarded as direct cues for enhanced prey accessibility through physical accumulation. The physical environment affects signal properties and without quantifying the different kinds of information that an animal can extract information from, it is challenging to obtain a mechanistic understanding of foraging behaviour [53,54]. Ultimately, investigating how an animal's perceptual abilities determine how it extracts information from the environment is an essential component of their foraging ability and thus, the animal's ecological function [55]. Therefore, our local-scale study can help formulate new hypotheses regarding sensory ecology [55], optimal foraging [56] and potential group dynamics [57] and collective motion [58].

In-flight terns must continually extract and process information from their environment which includes the visual challenge of locating an environmental cue at some distance which may be indicative of prey items. One of the central pieces of information that vision provides is the direction in which the target lies and the time it will take to arrive at the target (i.e. 'time-to-contact') while the actual distance to a target is of less importance [41]. This information is determined by optic flow which describes the way in which the image of the world moves across the retina as the head moves through space, which is essentially the perception of a non-uniform surface that changes continuously over time [41,59]. That birds use optical flow field information has been demonstrated in northern gannets *Morus bassanus* during plunge-diving manoeuvres [60]. Therefore, when a bird is lunging at an object, its movement and time-to-contact needs to be determined accurately [41]. It has been established that for most birds, distant prey is detected using lateral high-resolution vision, while at close range, the control of the bill close to the time of prey capture (including lunging) facilitates frontal/binocular vision [41]. For gull-billed terns *Gelochelidon nilotica*, there is evidence that they use lateral vision for the locating and tracking of potential prey [61]. The highly tortuous movements identified in our study during active foraging (state 1), also shows similarity to peregrine falcons *Falco peregrinus* that use curved paths to keep tracked prey in the central view of a single eye (lateral vision), before switching to binocular vision used for final prey capture [62]. In terns, the latter may be the case when its speed reduces to near-zero, indicating hovering, which often precedes a plunge dive at close range to the target.

The use of drones for optical sensing and tracking of surface flows using similar PIV techniques is now common practice [63,64]. Therefore, combining PIV methods with multi-target tracking is an attractive option when investigating ecological interactions. Our drone-based approach to quantify animal-environment interactions offers major advantages unobtainable with more traditional methods [65]. For instance, animal-borne telemetry applied to a few individuals may not capture movement within a specific area of interest if they do not frequent the site. Shore observations or vantage point surveys may quantify the relative number of birds using an area, but the oblique angle of the observer hinders the matching of a bird's spatial position to a feature underneath. Previously applied on bird colonies, drone enumerations have also been shown to be more precise than human counts [33,34,66]. While our approach can be applied to any surfacing marine vertebrate, seabirds pose a particular post-processing challenge. Seabirds in flight (compared to more static objects [67]) present a challenge for machine learning approaches, due to their small size, changing shape characteristics and especially when the spectral range of background turbulence is similar to that of the seabirds.

While our analytical approach explicitly acknowledges the time series nature of the observations, the relatively simple two-state HMM still is a strong simplification of the actual flight process observed at a very high resolution (30 Hz). In particular, figure 2d indicates strong momentum of both the speed and the log(tortuosity) also *within* either of the two HMM states. At the very fine (sub-second) scale considered, changes in speed and directionality are effectively continuous, such that the discretization into two states is indeed more plausible at a slightly coarser scale. This hierarchical structure of the variation in movement is not captured by our HMM, which assumes observations within the two discrete states to be conditionally independent. Including autoregressive terms in the observed process [68], or hierarchical model formulations that distinguish fine-scale and coarse-scale states [69] could improve the model's realism, but corresponding models are numerically much less stable, and are unlikely to give substantially different state classifications (which are highly plausible already when using our simpler approach). Alternatively, the very high serial correlation in the data could in principle be reduced by subsampling the time series at a lower resolution. Given the focus on highly localized scales, we preferred not to do this as to avoid any potential information loss. In the electronic supplementary material, figure S5, we do however show the results obtained when subsampling to 2.73 Hz (chosen to remove any additional correlation induced by the 11 element windows used for smoothing), and further an alternative analysis using the more common turning angles instead of log(tortuosity) to model the directional persistence. All of these approaches identified the same covariate effects. Overall, our methodological approach identified interesting correlations between behavioural modes and environmental cues, but did not explicitly model the choice of the target—this could for example be investigated using movement models with directional bias [70] or step-selection functions [71].

In conclusion, understanding how highly mobile marine predators extract information from their underlying environment may help us predict the potential impacts of environmental change [72]. This also concerns the introduction of man-made structures in our coastal seas [73], as these can influence the occurrence, scale and intensity of hydrodynamic features on local scales [74], thereby affecting foraging opportunities [6].

**Ethics.** No animals were sampled or approached. In order to minimize potential disturbance to foraging seabirds during the UAV hovers, the take-off and landing point of the UAV missions was chosen at a 200 m distance from at-sea foraging birds and the UAV was flown at 100 m above-sea level. Correspondence with the local department of environment (DAERA) prior to the study confirmed that no permits for the UAV surveys were necessary. The UAV surveys were

performed according to UK Civil Aviation Authority regulations and with the consent of the landowner for take-off and landing.

Data accessibility. The processed input dataset (.csv), including the tern tracking data and associated environmental data, and the R code (.txt) supporting this article's results (statistical analysis and figures) are available from the Dryad Digital Repository: https://doi.org/10.5061/dryad.kh189325b [75].

Authors' contributions. L.L. and W.A.M.N.-S. conceived the ideas and collected the data. All authors performed analyses and interpreted the results. L.L. drafted the initial manuscript. All authors contributed to writing and editing the final manuscript and gave the final approval for publication.

All authors gave final approval for publication and agreed to be held accountable for the work performed therein.

Competing interests. The authors declare no conflict of interest.

Funding. This study was funded by the Special EU Programmes Body (award number IVA5048).

Acknowledgements. We'd like to thank the Bryden Centre for advanced marine and bio-energy research. The Bryden Centre project is supported by the European Union's INTERREG VA Programme, managed by the Special EU Programmes Body (SEUPB). This research was inspired in part by the SFB TRR 212 (NC³), which is funded by the German Research Foundation (DFG). Finally, we thank the two anonymous reviewers who were tremendously helpful, providing constructive comments and suggestions which helped clarify and improve this manuscript.

Disclaimer. The views and opinions expressed in this paper do not necessarily reflect those of the European Commission or the Special EU Programmes Body (SEUPB).

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
