## [Peer Review File · Proceedings of the Royal Society B: Biological Sciences]

Review History

RSPB-2020-2881.R0 (Original submission)

Review form: Reviewer 1

Recommendation

Accept with minor revision (please list in comments)

Scientific importance: Is the manuscript an original and important contribution to its field?

Good

General interest: Is the paper of sufficient general interest?

Acceptable

Quality of the paper: Is the overall quality of the paper suitable?

Excellent

Is the length of the paper justified?

Yes

Should the paper be seen by a specialist statistical reviewer?

No

Do you have any concerns about statistical analyses in this paper? If so, please specify them explicitly in your report.

No

It is a condition of publication that authors make their supporting data, code and materials available - either as supplementary material or hosted in an external repository. Please rate, if applicable, the supporting data on the following criteria.

Is it accessible?

Yes

Is it clear?

Yes

Is it adequate?

Yes

Do you have any ethical concerns with this paper?

No

Comments to the Author

General comments

I very much enjoyed reading and reviewing this interesting and well-written manuscript, where the authors present a novel way of investigating the interactions between a surface-feeding seabird species and some ephemeral physical features generated by a man-made structure. I agree that the proposed technique offers an effective alternative to study such interactions at these fine scales, and that this exploration will allow us to ask very specific questions on how seabirds detect and select their feeding targets.

I don't have any expertise in image analysis, so I have to trust the procedure that was followed to extract the birds and the physical features from the videos. However, the rest of the analytical approach appears robust.

I have a few minor suggestions, which I detail below. I think they can mostly be addressed with some clarifications of the text and some additional details on the methods. The only point I am not entirely comfortable with is the extension of these results to all seabirds (or even all terns across habitats and times) in the Discussion. While the proposed technique could indeed be used for studying the feeding ecology of other species in other contexts, I don't think the results presented here are necessarily general (even though they do provide interesting insights for terns in this specific context). I would be careful not to oversell the findings in the Discussion.

Detailed comments

Line 28: 'vulnerable to coastal change' seems out of place here. You are discussing the issue of observational scale, and, while seabirds are certainly vulnerable to coastal change, this is disjunct from the rest of the sentence.

Line 30: other studies showed association with fronts, eddies and other mesoscale features (as you describe in the Introduction) - here, I think you are implying that it is the small spatial scale and extremely ephemeral nature of the features analysed that makes this study unique. I would be explicit. Also, there is no mention in the Abstract to the fact that these features were associated with a man-made structure - is it worth highlighting it?

Line 35: how is the tern a 'model' surface feeder? I would remove.

Line 37-38: this is vague: how do these results lay the foundation to understand coastal change? I would either be more specific or remove the sentence.

Line 74: I can't think of other volant marine predators- I would say either 'seabirds' or 'volant predators'.

Line 84: more common than what? I would just say 'common'.

Line 94: perceived by.

Line 115: 'occupying foraging behaviour' seems like a strange formulation; engaging in foraging behaviour?

Line 105-120: a minor comment: in this section, you alternatively refer to the study subjects as terns, birds, seabirds, animals, and surface-foraging marine predators, which makes it unclear to which group you are suggesting these results may apply. I would clarify.

Line 130: missing 'a' after 'generated'.

Line 141: is there a reason why hovers were 2-min long? I imagine longer hovers would have returned a larger sample size, or am I missing something?

Line 145: it might be useful to include a table with all hovers, and the corresponding times and number of tern tracks recorded.

Line 148: what did the calibration involve?

Line 170: what's the accuracy of the altimeter, and does it matter for subsequent analyses?

Line 172: how was this window size selected and does it affect the results?

Line 179-195: could you provide some references in support of this section? Particularly with regard to the corrections made and the standard functions used.

Line 198: because the coupling of bird and flow data is critical, I think this 'three-dimensional interpolation in space and time' requires some additional details.

Line 208: I would spell out 'absolute(curl)' for clarity.

Lines 228-231: is it worth describing model selection here?

Lines 239-240: I think this clarification should be moved to the Methods, where the two putative states are first mentioned (line 217?).

Lines 246-247: in general, I'm OK with the identification of those two discrete states. However, from Figure 2, it looks to me like there could be a more continuous change in speed and tortuosity, as reflected in the sample path shown and the not-really-bimodal state-dependent distributions. The option of fitting an alternative state space model with a continuous state may warrant a comment in the Discussion.

Lines 251-254: this sentence is a bit convoluted, I suggest some rephrasing; also, I would avoid repeating what was described in the Methods.

Line 272: again, I don't think you provide sufficient justification for why terns should be considered model surface foragers; I would expand on this point, or omit.

Line 282: do you feel comfortable extending your results to all seabirds? I can think of species that simply sit of the surface and feed from there, or species that use olfaction and fly much closer to the surface than terns. I imagine these species will show a different interaction with physical cues.

Line 285: a bit cryptic – is it worth explaining what these processes are?

Line 298: as per my comment above, I would be careful not to draw conclusions on all seabirds, which can move and hunt using very different strategies, from these results on terns studied in a peculiar context (and on a limited number of sampling occasions).

Line 306: what does 'its' refer to?

Line 307: formulate or formalise?

Lines 310-311: I don't understand this sentence, could you clarify?

Line 323: I would be consistent in how you use capital letters for species' common names.

Line 334: in the sense that tagged animals may not necessarily go through the area of interest? I would slightly rephrase to clarify.

Lines 335-337: the syntax of this sentence needs some revising.

Lines 353-356: I fully agree, but I am not sure this is a good/relevant concluding statement for this work.

Figure 4: I would include a legend for the colours in this (very nice!) plot.

Review form: Reviewer 2

Recommendation

Major revision is needed (please make suggestions in comments)

Scientific importance: Is the manuscript an original and important contribution to its field?

Good

General interest: Is the paper of sufficient general interest?

Good

Quality of the paper: Is the overall quality of the paper suitable?

Acceptable

Is the length of the paper justified?

Yes

Should the paper be seen by a specialist statistical reviewer?

No

Do you have any concerns about statistical analyses in this paper? If so, please specify them explicitly in your report.

No

It is a condition of publication that authors make their supporting data, code and materials available - either as supplementary material or hosted in an external repository. Please rate, if applicable, the supporting data on the following criteria.

Is it accessible?

Yes

Is it clear?

Yes

Is it adequate?

Yes

Do you have any ethical concerns with this paper?

No

Comments to the Author

Review: A bird's eye view on turbulence: Seabird foraging associations with evolving surface flow features

The authors present interesting research combining remarkably high detail tern tracking data and environmental data measured in situ using drones. Such analyses, if scaled to longer periods of time and more locations, could prove to be extremely valuable as the methods become increasingly accessible. Although the hypothesis and objectives were not clearly defined, the methods are adequate for presumed goal of the research; identifying effects of turbulent features of tern behaviour using drone-based telemetry. I first propose three moderate revisions followed by outlining smaller technical, grammatical, stylistic, and typographic revisions.

First, there needs to be a discussion and justify for the resolution that is used. The scale of the data determines the types of behaviours that can be identified. On one hand, coarse data may be prohibitive for studying finer-scale behaviours, however, at too fine a scale, different behaviours may begin to appear increasingly similar and be hard to differentiate. For example, at an extremely fine scale, the movement of nearly all behaviours appears perfectly straightened out. There are several factors that hint to me that the resolution may be too high for the desired behaviours. First, the use of tortuosity as a measure of turning (as opposed to turning angle), suggests that from point to point, movement is almost perfectly straightened out, and to detect state-specific differences in turning, it was necessary to calculate a turning metric across 11 locations (I also wonder whether this moving window for tortuosity violates the assumption of independent observation data in a basic HMM). Second, even with the use of tortuosity over turning angle, there is a very strong right skew of the tortuosity data in Fig. 2 B, which makes me suspect a lower resolution may exhibit a wider range of variation and make it easier to resolve distinct behaviours. Third, in Fig. 2 C there appear to be distinct flight paths at coarser scales may also be classified as "transit" and "active" foraging. It appears as though the current model is identifying individual turns from straightened section while there are sections of the path that are broad sweeping turns and sections of casting (i.e., "zig-zags"). Fourth, in Fig. 2 D it appears that there are many dozens of points within one bout of being in a behaviour, which may make it harder to identify the effects of covariates on the transition probabilities. This is supported by Fig 3 A and B, with the probabilities of remaining in the same state being exceptionally high (~0.99). The resolution that was used may indeed be valid for the behaviours investigated, however I suggest some more justification and discussion about the strengths and weaknesses of the resolution that was used. In addition, I would suggest including an appendix with the same analysis done on a lower resolution with corresponding versions of Fig. 2 and Fig 3, and a few sentences on similarities and differences. Such an appendix could be especially useful and insightful if future researchers are interested in investigating behaviours at slightly broader scales using similar ultra-high-resolution data, particularly if modelling directional bias or using step selection functions.

Second, I do not think that selecting the delays for the time-to-contact using likelihood necessarily tells us the scale at which the features affect behaviour. Instead, I think the likelihood approach

for delay selection tells us simply the scale at which most variation in the data is explained. However, stimuli likely have unique effects on behaviour that are different at different scales (i.e., different delays), and the effects (or lack thereof) at one delay provide uniquely important information from effects at another delay. For example, $\text{abs}(\text{curl})$ may have an ecologically significant effect on behaviour at a short time-scale as at a longer scale, however simply because the statistical likelihood is different, does not mean one is invalid. Therefore, I think there is merit to discussing the effects of vorticity and divergence at multiple delays, not only where the combination of delays produces the maximum likelihood. Especially since there are some very different delays that have a similar likelihood, for example, at a delay for $\text{abs}(\text{curl})$ of 0.25, a divergence delay of $d=0.5$ has comparable delta likelihood (-4.99) to a divergence delay of $d=4$ (-2.94); statistically relatively as powerful, but ecologically quite different. Specifically, I think it may be worth while to very briefly highlight 3 or 4 combinations of delays that have a decent likelihood but are far apart in table S1. It seems strange that upwelling ahead of the track cues behaviour to stay in transit, but does not cue to localised foraging when arrive at the source of upwelling.

Last, there are four smaller components that I think need to be more developed. First, the hypothesis and prediction should be stated more explicitly in the introduction and the discussion should link to these specifically. Presently, there is no hypothesis in the introduction, but a prediction that takes the form of a hypothesis and no true prediction. Furthermore, the discussion references a hypothesis not stated in the introduction and one that is not precisely linked to the prediction in the introduction. Second, the abstract and the discussion in general could benefit from more interpretation of the results, and the impact of findings to our ecological understanding of terns. Particularly because this research is framed in predicting responses to environmental change and unlocking knowledge gaps in seabird sensory foraging ecology. The interpretation of the results does not seem adequately tied back to these objectives. Does this add any new specific insights about this specific species or coastal feature? There are elements of this throughout the discussion but not a succinct focal point. Third, there is currently no discussion about limitations of the methods or results, and as a result, no specific suggestions for future research (either ecological or methodological). For example, I would have liked to see some discussion of other methods that might provide complimentary evidence of turbulent feature selection. For example, HMMs with directional bias toward targets, or using step selection analyses to explicitly examine selection, which HMMs cannot do as they do not consider available but unused habitat. Last, I think the authors should use the “state probability” or “stationary state probability” rather than “state occupancy”. “Occupancy” has a very specific definition with regard to animal movement/location data (see Lele et al. 2013. Selection, use, choice and occupancy: Clarifying concepts in resource selection studies.). Instead, I advise using “State probability” or “stationary state probability” (as the authors used in Fig. 3 caption), which would be more consistent with the terminology used in the HMM literature.

Specific feedback:

L24. Can remove “marine predator” for breadth or add “oceans” as on L43.

L25. Not self-evident that “turbulence” is referring to that of water and not air (especially since the title indicates the focal species is avian)

L26. Example of “turbulent features”?

L28. “vulnerable to coastal change” feels out of place.

L29. Specify the exact number of trajectories.

L30. “earliest evidence for predator associations with localised physical foraging cues” is extremely vague, and is certainly untrue. There is a copious amount of research on “predator associations with localised physical foraging cues”

L32. Indicate that this was based on several* species of terns? And consider noting the general region.

L34. “transit foraging” is undefined and not a colloquial behaviour.

L37. “understand [the effect of] coastal change [on...]”. The research does more than just understand coastal change.

L44-48. Consider switching second and third sentences

L48. “This” is vague

L49. Do such changes only generate new foraging opportunities? Or can they threaten them as

well?

L50. Last sentence feels long clunky. Streamline or split into two?

L61. Incorrect dash used

L68-69. Not clear why the examples are numbered. There are likely other physical processes that emerge in tidal environments, and you do not refer back to these two features explicitly.

L70-73. "some of these turbulent features..." This sentence describes how turbulent features specifically increase prey, which is first mentioned in the second sentence. could this sentence be moved up to the 3rd sentence?

L78. Specify the scale implied by "highly localised".

L78. Why the qualifier of the prey having to be "conspicuous"?

L82. Could "mechanisms underlying foraging strategies" be made more specific. I appreciate the specificity in the previous sentence, which concretely highlights "physical properties" to which seabirds "show affinity to".

L84. "such as [satellite-derived] animal telemetry [or animal tracking]". The drone-based approach in this research is technically "animal telemetry" (remote measure/monitoring of animal). Could also reword this part of the sentence to "such as associating coarse satellite-derived data with higher resolution animal tracking." This clarifies that the most common limitation is not necessarily the resolution of the track, but of associated environmental data, which is what the following sentence seems to be suggesting.

L84. "Animal telemetry" should not be hyphenated.

L86-86. It is the spatiotemporal mismatch that leads to not capturing highly localised associations.

Switch those two clauses. That is, something to the effect of "... averaged oceanographic data [results in a spatiotemporal mismatch...], and may obscure highly localised associations"

L93. Join with previous paragraph if starting with "specifically"

L94. Can remove "during foraging" as this can shed light on any number of behaviours

L96-97. A bird's eye view on underlying physical features does not quantify context-specific behaviours. It identifies features that can be subsequently used to identify context-specific behaviours. Reword.

L97. Remove space after thereby.

L101-104. Could you reframe this as a formal hypothesis, then add a specific prediction at the end of L113? E.g., Change L101 to "We [hypothesise] that surface-foraging terns (Sternidae) vary their foraging... which serve as physical cues [of high prey density] during foraging". The prediction should be the specific expected results of your analyses assuming your hypothesis is true. In this case, the prediction would be that the underlying turbulent features affect the state transition probabilities.

L103. "Physical cues of ____"

L114-116. As noted earlier in the introduction, other research has tied foraging to turbulent features at broader scales, presumably "specific" at their scale (e.g., reference 5). Reword to indicate that this is the earliest to do so at the sub-10-100m scale, or that this is the finest scale yet for remote-tracking-based data.

L120. This is not a sentence that should be referenced.

L148. What did the in-lab camera calibration entail? Consider removing if it was minimal.

L167. You use tortuosity here, but only define it on L174. The procedure should be defined for the first instance of its use.

L202-206. There is no need to introduce the "X" parameter, which is simply values of $d > 0$. It would be more concise to simply say "To investigate such 'time-to-contact' effects, time-offsets (delay $d \in \{0, 0.25, 0.5, \dots, 5\}$ in seconds) were applied, where $d = 0$ represents vorticity/divergence values extracted directly underneath the tern's position and $d > 0$ represent values ahead along the flight path"

L204. "X = 0.25 - 5 s" is misleading and suggests $X = -4.75$

L205-206. "To ensure parity between time-offsets, all tracks were truncated by the maximum offset of $d = 5$ s."

L212. Add space after 'travelling'

L216. Why did the authors model $\log(\text{tortuosity})$ and not simply tortuosity-1? Presumably both of these would yield values > 0 .

L216. Why do you use "abs(curl)" and "(log-) tortuosity" and not "log(tortuosity)"? I find "(log-)

tortuosity" confusing, it could be interpreted as $\log(-\text{tortuosity})$ (which produces complex numbers in this case), $-\log(\text{tortuosity})$, or $\log(\text{tortuosity})$. Alternatively, If you exclusively use $\text{abs}(\text{curl})$ and $\log(\text{tortuosity})$, you could clearly state this once and simply use "curl" and "tortuosity" thereafter.

L228. "d= 0-5.0" suggests $d = -5$. Either remove (as this has already been defined), or replace with "(i.e., delay $d \in \{0, 0.25, 0.5, \dots, 5\}$ s"

L228. Cite that you used the moveHMM package

L244. What is "opportunistically" referring to in the context of searching?

L246. Remove "clearly identifiable" as it is not objectively quantifiable

L247. Space after "Fig."

L251-252. Active foraging and transit foraging were previously defined as "state 1" and "state 2", which therefore should be the notation used (i.e., "Pr(state 1 -> state 2)", and "Pr(state 1)"). Or redefine states as TF and AF (transit foraging and active foraging, respectively).

L252. Here and throughout, I would advise against the use of "state occupancy" and instead use "state probability". "Occupancy" has a very specific definition in the field of movement ecology (see Lele et al. 2013. Selection, use, choice and occupancy: Clarifying concepts in resource selection studies.). "State probability", "stationary state distribution", or "stationary state probability" (as you have in Fig. 3 caption) would be more consistent with the terminology used in the HMM literature.

L258. would $\text{Pr}(i \rightarrow j)$ be more consistent with previous notation? 'X' was previously defined as $d > 0$.

L265. Consider: "Stationary state probability [Pr(i)]"

L274. "; d in seconds" is redundant with previous definitions

L279: "... [upwelling regions] do not necessarily evoke foraging activity": this has not been demonstrated in this research. See general comments about presenting results of $d = 0.25$ for upwelling.

L283. More specific on which local scales – "such localised scales" is ambiguous.

L289. This hypothesis was not previously articulated and is different from the prediction on line 101.

L308. Remove space after impacts.

L314. Replace "=" with "i.e.,"

L320. Do not start a paragraph with "therefore". Join with previous paragraph.

L343. Replace "e.g." with ", for example,"

L348. "while simultaneously [estimating] ecological...."

L554. Fig 2 C A starting location would be interesting to denote (particularly to compare with the time-series).

L599. 1 Fig 4 is not referenced in text and can be moved to the appendix.

L599. 2. Denote start location along track.

L599. 3 This is an interesting chart, but it is difficult to identify much meaningful patterns. The authors could indicate in the caption the main pattern that is being taken from the figure (e.g., transit toward vortex, followed by a transition to active foraging when surface feature is reached). It is quite hard to identify the height of surface features, particularly in mass on the left. Perhaps make the features much more translucent, but with outlines (with transparency approximately the same as the current areas).

Decision letter (RSPB-2020-2881.R0)

27-Jan-2021

Dear Dr Lieber:

I am writing to inform you that your manuscript RSPB-2020-2881 entitled "A bird's eye view on turbulence: Seabird foraging associations with evolving surface flow features" has, in its current form, been rejected for publication in Proceedings B.

This action has been taken on the advice of referees, who have recommended that substantial revisions are necessary. With this in mind we would be happy to consider a resubmission, provided the comments of the referees are fully addressed. However please note that this is not a provisional acceptance.

Sincerely,
 Professor Hans Heesterbeek
 mailto: proceedingsb@royalsociety.org

Associate Editor
 Board Member: 1
 Comments to Author:

Dear authors,

Two reviewers and myself have read the MS. We all see the potential of the MS and think it is an interesting study that is potentially suitable for the journal. Notwithstanding, both reviewers and myself have a list of points, ranging from relatively minor to quite major that we would like to see addressed. I think all of them are relevant and require serious consideration, but note that especially the reviewer's comment about the hypothesis in the introduction is important to address, and note myself that the results text is quite dense and technical and would like to have this made more accessible for a broader readership and more clearly linked textually to the biological questions being studied.

Specific comments AE:

1. Please make sure writing is accessible to a broad audience. For example, in the result one section is headed "HMM results on covariate effects". Preferably the heading focusses on the biological question being answered.
2. The last paragraph of the introduction already mentions results and what will be discussed, this is not Introduction material.

3. I found the section 5. Conclusion to be uninformative. All the statements are very generic, and could already be made before the start of the study (e.g. in the Introduction). It is too vague about how the results have furthered our understanding on this. Also please makes sure formatting is consistent with journal format (e.g. PRSB does not have a conclusion and background section).

Reviewer(s)' Comments to Author:

Referee: 1

Comments to the Author(s)

General comments

I very much enjoyed reading and reviewing this interesting and well-written manuscript, where the authors present a novel way of investigating the interactions between a surface-feeding seabird species and some ephemeral physical features generated by a man-made structure. I agree that the proposed technique offers an effective alternative to study such interactions at these fine scales, and that this exploration will allow us to ask very specific questions on how seabirds detect and select their feeding targets.

I don't have any expertise in image analysis, so I have to trust the procedure that was followed to extract the birds and the physical features from the videos. However, the rest of the analytical approach appears robust.

I have a few minor suggestions, which I detail below. I think they can mostly be addressed with some clarifications of the text and some additional details on the methods. The only point I am not entirely comfortable with is the extension of these results to all seabirds (or even all terns across habitats and times) in the Discussion. While the proposed technique could indeed be used for studying the feeding ecology of other species in other contexts, I don't think the results presented here are necessarily general (even though they do provide interesting insights for terns in this specific context). I would be careful not to oversell the findings in the Discussion.

Detailed comments

Line 28: 'vulnerable to coastal change' seems out of place here. You are discussing the issue of observational scale, and, while seabirds are certainly vulnerable to coastal change, this is disjunct from the rest of the sentence.

Line 30: other studies showed association with fronts, eddies and other mesoscale features (as you describe in the Introduction) – here, I think you are implying that it is the small spatial scale and extremely ephemeral nature of the features analysed that makes this study unique. I would be explicit. Also, there is no mention in the Abstract to the fact that these features were associated with a man-made structure – is it worth highlighting it?

Line 35: how is the tern a 'model' surface feeder? I would remove.

Line 37-38: this is vague: how do these results lay the foundation to understand coastal change? I would either be more specific or remove the sentence.

Line 74: I can't think of other volant marine predators– I would say either 'seabirds' or 'volant predators'.

Line 84: more common than what? I would just say 'common'.

Line 94: perceived by.

Line 115: 'occupying foraging behaviour' seems like a strange formulation; engaging in foraging behaviour?

Line 105-120: a minor comment: in this section, you alternatively refer to the study subjects as terns, birds, seabirds, animals, and surface-foraging marine predators, which makes it unclear to which group you are suggesting these results may apply. I would clarify.

Line 130: missing 'a' after 'generated'.

Line 141: is there a reason why hovers were 2-min long? I imagine longer hovers would have returned a larger sample size, or am I missing something?

Line 145: it might be useful to include a table with all hovers, and the corresponding times and number of tern tracks recorded.

Line 148: what did the calibration involve?

Line 170: what's the accuracy of the altimeter, and does it matter for subsequent analyses?

Line 172: how was this window size selected and does it affect the results?

Line 179-195: could you provide some references in support of this section? Particularly with regard to the corrections made and the standard functions used.

Line 198: because the coupling of bird and flow data is critical, I think this 'three-dimensional interpolation in space and time' requires some additional details.

Line 208: I would spell out 'absolute(curl)' for clarity.

Lines 228-231: is it worth describing model selection here?

Lines 239-240: I think this clarification should be moved to the Methods, where the two putative states are first mentioned (line 217?).

Lines 246-247: in general, I'm OK with the identification of those two discrete states. However, from Figure 2, it looks to me like there could be a more continuous change in speed and tortuosity, as reflected in the sample path shown and the not-really-bimodal state-dependent distributions. The option of fitting an alternative state space model with a continuous state may warrant a comment in the Discussion.

Lines 251-254: this sentence is a bit convoluted, I suggest some rephrasing; also, I would avoid repeating what was described in the Methods.

Line 272: again, I don't think you provide sufficient justification for why terns should be considered model surface foragers; I would expand on this point, or omit.

Line 282: do you feel comfortable extending your results to all seabirds? I can think of species that simply sit of the surface and feed from there, or species that use olfaction and fly much closer to the surface than terns. I imagine these species will show a different interaction with physical cues.

Line 285: a bit cryptic – is it worth explaining what these processes are?

Line 298: as per my comment above, I would be careful not to draw conclusions on all seabirds, which can move and hunt using very different strategies, from these results on terns studied in a peculiar context (and on a limited number of sampling occasions).

Line 306: what does 'its' refer to?

Line 307: formulate or formalise?

Lines 310-311: I don't understand this sentence, could you clarify?

Line 323: I would be consistent in how you use capital letters for species' common names.

Line 334: in the sense that tagged animals may not necessarily go through the area of interest? I would slightly rephrase to clarify.

Lines 335-337: the syntax of this sentence needs some revising.

Lines 353-356: I fully agree, but I am not sure this is a good/relevant concluding statement for this work.

Figure 4: I would include a legend for the colours in this (very nice!) plot.

Referee: 2

Comments to the Author(s)

Review: A bird's eye view on turbulence: Seabird foraging associations with evolving surface flow features

The authors present interesting research combining remarkably high detail tern tracking data and environmental data measured in situ using drones. Such analyses, if scaled to longer periods of time and more locations, could prove to be extremely valuable as the methods become increasingly accessible. Although the hypothesis and objectives were not clearly defined, the methods are adequate for presumed goal of the research; identifying effects of turbulent features of tern behaviour using drone-based telemetry. I first propose three moderate revisions followed by outlining smaller technical, grammatical, stylistic, and typographic revisions.

First, there needs to be a discussion and justify for the resolution that is used. The scale of the data determines the types of behaviours that can be identified. On one hand, coarse data may be prohibitive for studying finer-scale behaviours, however, at too fine a scale, different behaviours may begin to appear increasingly similar and be hard to differentiate. For example, at an extremely fine scale, the movement of nearly all behaviours appears perfectly straightened out. There are several factors that hint to me that the resolution may be too high for the desired behaviours. First, the use of tortuosity as a measure of turning (as opposed to turning angle), suggests that from point to point, movement is almost perfectly straightened out, and to detect state-specific differences in turning, it was necessary to calculate a turning metric across 11 locations (I also wonder whether this moving window for tortuosity violates the assumption of independent observation data in a basic HMM). Second, even with the use of tortuosity over turning angle, there is a very strong right skew of the tortuosity data in Fig. 2 B, which makes me suspect a lower resolution may exhibit a wider range of variation and make it easier to resolve distinct behaviours. Third, in Fig. 2 C there appear to be distinct flight paths at coarser scales may also be classified as "transit" and "active" foraging. It appears as though the current model is identifying individual turns from straightened section while there are sections of the path that are broad sweeping turns and sections of casting (i.e., "zig-zags"). Fourth, in Fig. 2 D it appears that there are many dozens of points within one bout of being in a behaviour, which may make it harder to identify the effects of covariates on the transition probabilities. This is supported by Fig 3 A and B, with the probabilities of remaining in the same state being exceptionally high (~0.99). The resolution that was used may indeed be valid for the behaviours investigated, however I suggest some more justification and discussion about the strengths and weaknesses of the resolution that was used. In addition, I would suggest including an appendix with the same analysis done on a lower resolution with corresponding versions of Fig. 2 and Fig 3, and a few sentences on similarities and differences. Such an appendix could be especially useful and

insightful if future researchers are interested in investigating behaviours at slightly broader scales using similar ultra-high-resolution data, particularly if modelling directional bias or using step selection functions.

Second, I do not think that selecting the delays for the time-to-contact using likelihood necessarily tells us the scale at which the features affect behaviour. Instead, I think the likelihood approach for delay selection tells us simply the scale at which most variation in the data is explained.

However, stimuli likely have unique effects on behaviour that are different at different scales (i.e., different delays), and the effects (or lack thereof) at one delay provide uniquely important information from effects at another delay. For example, $\text{abs}(\text{curl})$ may have an ecologically significant effect on behaviour at a short time-scale as at a longer scale, however simply because the statistical likelihood is different, does not mean one is invalid. Therefore, I think there is merit to discussing the effects of vorticity and divergence at multiple delays, not only where the combination of delays produces the maximum likelihood. Especially since there are some very different delays that have a similar likelihood, for example, at a delay for $\text{abs}(\text{curl})$ of 0.25, a divergence delay of $d=0.5$ has comparable delta likelihood (-4.99) to a divergence delay of $d=4$ (-2.94); statistically relatively as powerful, but ecologically quite different. Specifically, I think it may be worth while to very briefly highlight 3 or 4 combinations of delays that have a decent likelihood but are far apart in table S1. It seems strange that upwelling ahead of the track cues behaviour to stay in transit, but does not cue to localised foraging when arrive at the source of upwelling.

Last, there are four smaller components that I think need to be more developed. First, the hypothesis and prediction should be stated more explicitly in the introduction and the discussion should link to these specifically. Presently, there is no hypothesis in the introduction, but a prediction that takes the form of a hypothesis and no true prediction. Furthermore, the discussion references a hypothesis not stated in the introduction and one that is not precisely linked to the prediction in the introduction. Second, the abstract and the discussion in general could benefit from more interpretation of the results, and the impact of findings to our ecological understanding of terns. Particularly because this research is framed in predicting responses to environmental change and unlocking knowledge gaps in seabird sensory foraging ecology. The interpretation of the results does not seem adequately tied back to these objectives. Does this add any new specific insights about this specific species or coastal feature? There are elements of this throughout the discussion but not a succinct focal point. Third, there is currently no discussion about limitations of the methods or results, and as a result, no specific suggestions for future research (either ecological or methodological). For example, I would have liked to see some discussion of other methods that might provide complimentary evidence of turbulent feature selection. For example, HMMs with directional bias toward targets, or using step selection analyses to explicitly examine selection, which HMMs cannot do as they do not consider available but unused habitat. Last, I think the authors should use the “state probability” or “stationary state probability” rather than “state occupancy”. “Occupancy” has a very specific definition with regard to animal movement/location data (see Lele et al. 2013. Selection, use, choice and occupancy: Clarifying concepts in resource selection studies.). Instead, I advise using “State probability” or “stationary state probability” (as the authors used in Fig. 3 caption), which would be more consistent with the terminology used in the HMM literature.

Specific feedback:

L24. Can remove “marine predator” for breadth or add “oceans” as on L43.

L25. Not self-evident that “turbulence” is referring to that of water and not air (especially since the title indicates the focal species is avian)

L26. Example of “turbulent features”?

L28. “vulnerable to coastal change” feels out of place.

L29. Specify the exact number of trajectories.

L30. “earliest evidence for predator associations with localised physical foraging cues” is extremely vague, and is certainly untrue. There is a copious amount of research on “predator associations with localised physical foraging cues”

L32. Indicate that this was based on several* species of terns? And consider noting the general region.

L34. “transit foraging” is undefined and not a colloquial behaviour.

L37. “understand [the effect of] coastal change [on...]”. The research does more than just understand coastal change.

L44-48. Consider switching second and third sentences

L48. “This” is vague

L49. Do such changes only generate new foraging opportunities? Or can they threaten them as well?

L50. Last sentence feels long clunky. Streamline or split into two?

L61. Incorrect dash used

L68-69. Not clear why the examples are numbered. There are likely other physical processes that emerge in tidal environments, and you do not refer back to these two features explicitly.

L70-73. “some of these turbulent features...” This sentence describes how turbulent features specifically increase prey, which is first mentioned in the second sentence. could this sentence be moved up to the 3rd sentence?

L78. Specify the scale implied by “highly localised”.

L78. Why the qualifier of the prey having to be “conspicuous”?

L82. Could “mechanisms underlying foraging strategies” be made more specific. I appreciate the specificity in the previous sentence, which concretely highlights “physical properties” to which seabirds “show affinity to”.

L84. “such as [satellite-derived] animal telemetry [or animal tracking]”. The drone-based approach in this research is technically “animal telemetry” (remote measure/monitoring of animal). Could also reword this part of the sentence to “such as associating coarse satellite-derived data with higher resolution animal tracking.” This clarifies that the most common limitation is not necessarily the resolution of the track, but of associated environmental data, which is what the following sentence seems to be suggesting.

L84. “Animal telemetry” should not be hyphenated.

L86-86. It is the spatiotemporal mismatch that leads to not capturing highly localised associations. Switch those two clauses. That is, something to the effect of “... averaged oceanographic data [results in a spatiotemporal mismatch...], and may obscure highly localised associations”

L93. Join with previous paragraph if starting with “specifically”

L94. Can remove “during foraging” as this can shed light on any number of behaviours

L96-97. A bird’s eye view on underlying physical features does not quantify context-specific behaviours. It identifies features that can be subsequently used to identify context-specific behaviours. Reword.

L97. Remove space after thereby.

L101-104. Could you reframe this as a formal hypothesis, then add a specific prediction at the end of L113? E.g., Change L101 to “We [hypothesise] that surface-foraging terns (Sternidae) vary their foraging... which serve as physical cues [of high prey density] during foraging”. The prediction should be the specific expected results of your analyses assuming your hypothesis is true. In this case, the prediction would be that the underlying turbulent features affect the state transition probabilities.

L103. “Physical cues of_____”

L114-116. As noted earlier in the introduction, other research has tied foraging to turbulent features at broader scales, presumably “specific” at their scale (e.g., reference 5). Reword to indicate that this is the earliest to do so at the sub-10-100m scale, or that this is the finest scale yet for remote-tracking-based data.

L120. This is not a sentence that should be referenced.

L148. What did the in-lab camera calibration entail? Consider removing if it was minimal.

L167. You use tortuosity here, but only define it on L174. The procedure should be defined for the first instance of its use.

L202-206. There is no need to introduce the “X” parameter, which is simply values of $d > 0$. It would be more concise to simply say “To investigate such ‘time-to-contact’ effects, time-offsets (delay $d \in \{0, 0.25, 0.5, \dots, 5\}$ in seconds) were applied, where $d = 0$ represents vorticity/divergence values extracted directly underneath the tern’s position and $d > 0$ represent values ahead along the flight path”

L204. “ $X = 0.25 - 5 \text{ s}$ ” is misleading and suggests $X = -4.75$

L205-206. "To ensure parity between time-offsets, all tracks were truncated by the maximum offset of $d = 5$ s."

L212. Add space after 'travelling'

L216. Why did the authors model $\log(\text{tortuosity})$ and not simply $\text{tortuosity}-1$? Presumably both of these would yield values >0 .

L216. Why do you use " $\text{abs}(\text{curl})$ " and " $(\log-)$ tortuosity" and not " $\log(\text{tortuosity})$ "? I find " $(\log-)$ tortuosity" confusing, it could be interpreted as $\log(-\text{tortuosity})$ (which produces complex numbers in this case), $-\log(\text{tortuosity})$, or $\log(\text{tortuosity})$. Alternatively, If you exclusively use $\text{abs}(\text{curl})$ and $\log(\text{tortuosity})$, you could clearly state this once and simply use " curl " and " tortuosity " thereafter.

L228. " $d = 0-5.0$ " suggests $d = -5$. Either remove (as this has already been defined), or replace with "(i.e., delay $d \in \{0, 0.25, 0.5, \dots, 5\}$ s"

L228. Cite that you used the `moveHMM` package

L244. What is "opportunistically" referring to in the context of searching?

L246. Remove "clearly identifiable" as it is not objectively quantifiable

L247. Space after "Fig."

L251-252. Active foraging and transit foraging were previously defined as "state 1" and "state 2", which therefore should be the notation used (i.e., " $\text{Pr}(\text{state 1} \rightarrow \text{state 2})$ ", and " $\text{Pr}(\text{state 1})$ "). Or redefine states as TF and AF (transit foraging and active foraging, respectively).

L252. Here and throughout, I would advise against the use of "state occupancy" and instead use "state probability". "Occupancy" has a very specific definition in the field of movement ecology (see Lele et al. 2013. Selection, use, choice and occupancy: Clarifying concepts in resource selection studies.). "State probability", "stationary state distribution", or "stationary state probability" (as you have in Fig. 3 caption) would be more consistent with the terminology used in the HMM literature.

L258. would $\text{Pr}(i \rightarrow j)$ be more consistent with previous notation? 'X' was previously defined as $d > 0$.

L265. Consider: "Stationary state probability [$\text{Pr}(i)$]"

L274. "; d in seconds" is redundant with previous definitions

L279: "... [upwelling regions] do not necessarily evoke foraging activity": this has not been demonstrated in this research. See general comments about presenting results of $d = 0.25$ for upwelling.

L283. More specific on which local scales - "such localised scales" is ambiguous.

L289. This hypothesis was not previously articulated and is different from the prediction on line 101.

L308. Remove space after impacts.

L314. Replace "=" with "i.e.,"

L320. Do not start a paragraph with "therefore". Join with previous paragraph.

L343. Replace "e.g." with ", for example,"

L348. "while simultaneously [estimating] ecological...."

L554. Fig 2 C A starting location would be interesting to denote (particularly to compare with the time-series).

L599. 1 Fig 4 is not referenced in text and can be moved to the appendix.

L599. 2. Denote start location along track.

L599. 3 This is an interesting chart, but it is difficult to identify much meaningful patterns. The authors could indicate in the caption the main pattern that is being taken from the figure (e.g., transit toward vortex, followed by a transition to active foraging when surface feature is reached). It is quite hard to identify the height of surface features, particularly in mass on the left. Perhaps make the features much more translucent, but with outlines (with transparency approximately the same as the current areas).

Author's Response to Decision Letter for (RSPB-2020-2881.R0)

See Appendix A.

RSPB-2021-0592.R0

Review form: Reviewer 2

Recommendation

Accept with minor revision (please list in comments)

Scientific importance: Is the manuscript an original and important contribution to its field?

Excellent

General interest: Is the paper of sufficient general interest?

Excellent

Quality of the paper: Is the overall quality of the paper suitable?

Good

Is the length of the paper justified?

Yes

Should the paper be seen by a specialist statistical reviewer?

No

Do you have any concerns about statistical analyses in this paper? If so, please specify them explicitly in your report.

No

It is a condition of publication that authors make their supporting data, code and materials available - either as supplementary material or hosted in an external repository. Please rate, if applicable, the supporting data on the following criteria.

Is it accessible?

N/A

Is it clear?

N/A

Is it adequate?

N/A

Do you have any ethical concerns with this paper?

No

Comments to the Author

General Comments:

I thank the authors for taking time to address all of the points raised in my first review and I believe the revised manuscript is significantly improved! I am satisfied with the response to my review, and I particularly appreciate the additional methods investigating a lower resolution and the use of turning angle. Although it did not change the results, it eliminates a potentially

significant point of uncertainty and provides a starting point for future analyses. If data and R code is provided as supplementary material (which I do not currently see), you should note this somewhere in the manuscript. I support publication of the manuscript following minor grammatical revisions listed below.

Specific comments:

LL48-51. Sentence is a bit of a run on. Perhaps new sentence after “predators and installations” and maybe start with “There is some evidence that installations can..., however we are yet to understand how...”

LL66. Remove comma before “as a result of”

LL66. Consider moving the citations to the end of sentence.

LL67. Add a comma before “or physical aggregation...” (as on LL 92 and elsewhere in the paper)

LL73. Remove comma before “and dynamic boundary...”

LL98. Add quotes (I think single) around “bird’s eye view”

LL103/112. Much improved hypothesis/prediction. Thank you.

LL112. Remove “, respectively”

LL270. “ $d=0.25/2.0$ ” suggests $d = 0.125$. Either clarify with something along the lines of “ $d=0.25$ for curl and $d=0.5$ for divergence” or remove all together as it was mentioned three sentences prior.

LL270-272. Make the brackets into a complete sentence and add a reference to the supplementary material table showing this.

LL287-291. Good H/P.

LL326. Did you mean “sensory ecology” rather than “sensor ecology”?

LL328. Can remove brackets around “through physical accumulation”.

LL371-392. Excellent addition to discussion and gives lots to think about!

Decision letter (RSPB-2021-0592.R0)

26-Mar-2021

Dear Dr Lieber:

Your manuscript has now been peer reviewed and the review has been assessed by an Associate Editor. The reviewer's comments (not including confidential comments to the Editor) and the comments from the Associate Editor are included at the end of this email for your reference. As you will see, the reviewer and the Associate Editor are positive about the revision. They have raised some issues with your manuscript and we would like to invite you to revise your manuscript to address them. Note in particular that it is a condition for publication that data are made available and that this currently does not seem to be the case. We cannot proceed until this issue is solved.

When submitting your revision please upload a file under "Response to Referees" in the "File Upload" section. This should document, point by point, how you have responded to the

reviewers' and Editors' comments, and the adjustments you have made to the manuscript. We require a copy of the manuscript with revisions made since the previous version marked as 'tracked changes' to be included in the 'response to referees' document.

Research ethics:

Use of animals and field studies:

It is a condition of publication that you make available the data and research materials supporting the results in the article (<https://royalsociety.org/journals/authors/author-guidelines/#data>). Datasets should be deposited in an appropriate publicly available repository and details of the associated accession number, link or DOI to the datasets must be included in the Data Accessibility section of the article (<https://royalsociety.org/journals/ethics-policies/data-sharing-mining/>). Reference(s) to datasets should also be included in the reference list of the article with DOIs (where available).

Online supplementary material will also carry the title and description provided during submission, so please ensure these are accurate and informative. Note that the Royal Society will not edit or typeset supplementary material and it will be hosted as provided. Please ensure that

the supplementary material includes the paper details (authors, title, journal name, article DOI). Your article DOI will be 10.1098/rspb.[paper ID in form xxxx.xxxx e.g. 10.1098/rspb.2016.0049].

Please submit a copy of your revised paper within three weeks. If we do not hear from you within this time your manuscript will be rejected. If you are unable to meet this deadline please let us know as soon as possible, as we may be able to grant a short extension.

Best wishes,
Professor Hans Heesterbeek
mailto: proceedingsb@royalsociety.org

Associate Editor Board Member

Comments to Author:

Dear authors,

One of the original reviewers and myself have read your revision, and we are both happy with the rigorous way how you have dealt with the previous comments. Some relatively minor points remain about the accessibility of code and data and some grammatical points raised by the reviewer.

Reviewer(s)' Comments to Author:

Referee: 2

Comments to the Author(s).

General Comments:

I thank the authors for taking time to address all of the points raised in my first review and I believe the revised manuscript is significantly improved! I am satisfied with the response to my review, and I particularly appreciate the additional methods investigating a lower resolution and the use of turning angle. Although it did not change the results, it eliminates a potentially significant point of uncertainty and provides a starting point for future analyses. If data and R code is provided as supplementary material (which I do not currently see), you should note this somewhere in the manuscript. I support publication of the manuscript following minor grammatical revisions listed below.

Specific comments:

LL48-51. Sentence is a bit of a run on. Perhaps new sentence after “predators and installations” and maybe start with “There is some evidence that installations can..., however we are yet to understand how...”

LL66. Remove comma before “as a result of”

LL66. Consider moving the citations to the end of sentence.

LL67. Add a comma before “or physical aggregation...” (as on LL 92 and elsewhere in the paper)

LL73. Remove comma before “and dynamic boundary...”

LL98. Add quotes (I think single) around “bird’s eye view”

LL103/112. Much improved hypothesis/prediction. Thank you.

LL112. Remove “, respectively”

LL270. “ $d=0.25/2.0$ ” suggests $d = 0.125$. Either clarify with something along the lines of “ $d=0.25$ for curl and $d=0.5$ for divergence” or remove all together as it was mentioned three sentences prior.

LL270-272. Make the brackets into a complete sentence and add a reference to the supplementary material table showing this.

LL287-291. Good H/P.

LL326. Did you mean “sensory ecology” rather than “sensor ecology”?

LL328. Can remove brackets around “through physical accumulation”.

LL371-392. Excellent addition to discussion and gives lots to think about!

Author's Response to Decision Letter for (RSPB-2021-0592.R0)

See Appendix B.

Decision letter (RSPB-2021-0592.R1)

01-Apr-2021

Dear Dr Lieber

I am pleased to inform you that your manuscript entitled "A bird's eye view on turbulence: Seabird foraging associations with evolving surface flow features" has been accepted for publication in Proceedings B.

Data Accessibility section

Open Access

Paper charges

Sincerely,
Professor Hans Heesterbeek
Editor, Proceedings B
mailto: proceedingsb@royalsociety.org

Associate Editor:
Board Member
Comments to Author:
(There are no comments.)

Appendix A

QUEEN'S
UNIVERSITY
BELFAST

THE
BRYDEN
CENTRE

ADVANCED MARINE AND BIO-ENERGY RESEARCH

Dr Lilian Lieber
Queen's University Belfast, Marine Laboratory
12-13 The Strand
Portaferry, BT22 1PF
Northern Ireland
Tel: 0044 (0) 7837425855
E-mail: l.lieber@qub.ac.uk

Editor
Proceedings of the Royal Society B

March 3rd, 2021

Dear Professor Hans Heesterbeek,

We are pleased to re-submit our Article entitled "A bird's eye view on turbulence: Seabird foraging associations with evolving surface flow features" (MS Reference Number: RSPB-2020-2881) by Lieber *et al.* for consideration for publication in *Proceedings of the Royal Society B*.

We appreciate the previous consideration of our manuscript and have now fully addressed the helpful comments made by yourself and the reviewers. We have uploaded a full and complete 'response to referees' document, as well as the clean and tracked-changes versions of the manuscript.

The key changes are, that we have re-formulated and clarified the study's hypothesis and prediction to be more specific and have made changes throughout the Discussion accordingly. This includes a more thorough interpretation of our results as requested. As suggested by Reviewer 2, we have now run our models with a different track resolution as well as with different movement metrics to decode the states. This identified the same covariate effects. We have now included discussion on this, and provide additional model outputs in the supplementary material.

There is a high interest in 'high-throughput movement ecology' at the moment, and we believe our work will make a valuable contribution to this emerging field.

Our findings provide the clearest evidence to date that seabirds change their foraging strategies in relation to physical cues on local scales. Without an understanding of the physical drivers of foraging, we cannot predict seabird vulnerability to coastal change, such as the introduction of anthropogenic installations which influence the occurrence, scale and intensity of localised hydrodynamic features.

This manuscript describes original work and is not under consideration by any other journal. All authors approved the manuscript and this resubmission. Thank you for your consideration of our manuscript. We look forward to hearing your response.

Sincerely,

Lilian Lieber, PhD
Queen's University Belfast

Associate Editor

Board Member: 1

Comments to Author:

Dear authors,

Two reviewers and myself have read the MS. We all see the potential of the MS and think it is an interesting study that is potentially suitable for the journal. Notwithstanding, both reviewers and myself have a list of points, ranging from relatively minor to quite major that we would like to see addressed. I think all of them are relevant and require serious consideration, but note that especially the reviewer's comment about the hypothesis in the introduction is important to address, and note myself that the results text is quite dense and technical and would like to have this made more accessible for a broader readership and more clearly linked textually to the biological questions being studied.

We thank the Editor and the reviewers for the thoughtful consideration of our manuscript and the helpful comments. We have re-formulated the study's hypothesis and prediction to be more specific and have made changes throughout the Discussion accordingly, including a more thorough interpretation of our results. We also appreciate the more major points which we were able to address accordingly. As suggested by Reviewer 2, we have now run our models with a different track resolution as well as with different movement metrics to decode the states, namely changing tortuosity to turning angles – the additional analysis is now included in the supplementary material and shows that the state-dependent distributions as well as associated transition probabilities do not differ from the original results. We have added text to the Discussion detailing the additional analysis and the model approach used. There is a high interest in "high-throughput movement ecology" at the moment, and we believe our work will make a valuable contribution to this emerging field.

Specific comments AE:

1. Please make sure writing is accessible to a broad audience. For example, in the result one section is headed "HMM results on covariate effects".

Preferably the heading focusses on the biological question being answered.

We appreciate this comment and have changed the wording and section headings in the Results to be more comprehensive to a broad audience, e.g.: 'Effects of turbulence features on foraging states'.

2. The last paragraph of the introduction already mentions results and what will be discussed, this is not Introduction material.

We have re-phrased this final paragraph of the introduction and have deleted the final sentence on the discussion.

3. I found the section 5. Conclusion to be uninformative. All the statements are very generic, and could already be made before the start of the study (e.g. in the Introduction). It is too vague about how the results have furthered our understanding on this. Also please makes sure formatting is consistent with journal format (e.g. PRSB does not have a conclusion and background section).

We have changed 'Background' to 'Introduction'. Further, we have now completely rewritten this final section. Please note the substantial changes made in the second paragraph of the discussion, where we have now added more interpretation of our results to put our work into context, while indicating future hypothesis testing.

Finally, please note, that all changes below indicating line numbers now refer to the new, clean version of the manuscript.

Reviewer(s)' Comments to Author:

Referee: 1

Comments to the Author(s)

General comments

I very much enjoyed reading and reviewing this interesting and well-written manuscript, where the authors present a novel way of investigating the interactions between a surface-feeding seabird species and some ephemeral physical features generated by a man-made structure. I agree that the proposed technique offers an effective alternative to study such interactions at these fine scales, and that this exploration will allow us to ask very specific questions on how seabirds detect and select their feeding targets.

I don't have any expertise in image analysis, so I have to trust the procedure that was followed to extract the birds and the physical features from the videos. However, the rest of the analytical approach appears robust.

I have a few minor suggestions, which I detail below. I think they can mostly be addressed with some clarifications of the text and some additional details on the methods. The only point I am not entirely comfortable with is the

extension of these results to all seabirds (or even all terns across habitats and times) in the Discussion. While the proposed technique could indeed be used for studying the feeding ecology of other species in other contexts, I don't think the results presented here are necessarily general (even though they do provide interesting insights for terns in this specific context). I would be careful not to oversell the findings in the Discussion.

We'd like to thank the reviewer for the support and thoughtful consideration of our manuscript. We have addressed all comments made by the reviewer. There is increasing evidence emerging that terns species focus their foraging activity over areas of fast flows and turbulence (Schwemmer et al. 2009, Urmy & Warren 2018, Lieber et al 2019), detailing the importance of small-scale physical processes directly influencing prey availability. Our study provides a novel approach to contextualise and gain first insights into the importance of underlying physical foraging cues on scales that were not achieved previously. In accordance with the observations from the referenced studies above and our own observations at similar sites (in and outside of the tidal channel), we have confidence that this is not a site-specific study as turbulence features as the ones described also exist, for example, around natural whirlpool features or wake areas where terns are often to be found foraging. However, we have made changes in the Discussion to reflect when and where our presented results may apply and hope to continue this exciting research across diverse sites in the future.

Detailed comments

Line 28: 'vulnerable to coastal change' seems out of place here. You are discussing the issue of observational scale, and, while seabirds are certainly vulnerable to coastal change, this is disjunct from the rest of the sentence.

We have removed 'vulnerable to coastal change'. Please note, this originally referred to the first sentence of the abstract; 'predicting responses to environmental change'. Our ability to predict responses to environmental change, such as the introduction of man-made structures in dynamic coastal environments, is limited unless we develop methods to understand seabird-resource interactions.

Line 30: other studies showed association with fronts, eddies and other mesoscale features (as you describe in the Introduction) – here, I think you are implying that it is the small spatial scale and extremely ephemeral nature of the features analysed that makes this study unique. I would be explicit. Also, there is no mention in the Abstract to the fact that these features were associated with a man-made structure – is it worth highlighting it?

Yes, precisely. Other studies have found foraging associations with larger-scale (e.g. meso-scale; in the order of 10-100s Km) features, whereas this is the first study to investigate seabird associations with localised features in the order of 10s-100s m. We appreciate that this may have been unclear, and have now added scale for clarity, L26:

'Yet, identifying ecologically important localised turbulence features (e.g. upwellings ~10-100 m) is limited by observational scale and this knowledge gap is magnified in volatile predators.'

The turbulent features under consideration are characteristic of high-flow environments and therefore more broadly applicable. As such, they are representative for turbulence arising from either, man-made or natural wake structures. We have now clarified this in the second sentence of the abstract, L25:

'For instance, turbulence in the water arising from natural or anthropogenic structures can affect foraging opportunities in tidal seas.'

Line 35: how is the tern a 'model' surface feeder? I would remove.

We agree and have removed 'model'. We used the word 'model' to underpin that the terns in this study are representative of other tern species (e.g. showing similarities to tern foraging behaviour investigated in Urmy & Warren 2018) and indeed, other surface-feeding seabirds that rely on physical mechanisms to bring prey near the sea surface.

Urmy SS, Warren JD. 2018 Foraging hotspots of common and roseate terns: the influence of tidal currents, bathymetry, and prey density. *Mar. Ecol. Prog. Ser.* **590**, 227–245

Line 37-38: this is vague: how do these results lay the foundation to understand coastal change? I would either be more specific or remove the sentence.

This is in the context of anthropogenic development in the marine environment ('marine urbanisation') and associated coastal change. We argue that in order to predict responses of e.g. seabirds to anthropogenic installations (which result in environmental change, specifically changes in local flow conditions), we need to understand the physical mechanisms underlying foraging behaviour (as this will affect foraging success – an important parameter to understand population-level changes). We have now added more context in the previous sentences, and hope that it becomes more clear to the reader, having done the following modification, L37:

'Finally, it lays the foundation to predict responses to coastal change to inform sustainable ocean development.'

Line 74: I can't think of other volant marine predators– I would say either 'seabirds' or 'volant predators'.

We agree and have rewritten it to: 'Volant predators'.

Line 84: more common than what? I would just say 'common'.

We have removed 'more' and changed 'common' to 'traditional'.

Line 94: perceived by.

We have changed this accordingly.

Line 115: 'occupying foraging behaviour' seems like a strange formulation; engaging in foraging behaviour?

We agree and due to additional comments from the other reviewers, we have now re-written the last paragraph of the introduction (L113):

'This allowed us to quantify the influence of prevalent oceanographic features on a surface-foraging marine predator on hitherto unobtainable scales (~10 –100 m).'

Line 105-120: a minor comment: in this section, you alternatively refer to the study subjects as terns, birds, seabirds, animals, and surface-foraging marine predators, which makes it unclear to which group you are suggesting these results may apply. I would clarify.

We agree and re-worded some terms to be more consistent (i.e. changing 'birds' to 'terns').

Line 130: missing 'a' after 'generated'.

Thank you, we have inserted 'a'.

Line 141: is there a reason why hovers were 2-min long? I imagine longer hovers would have returned a larger sample size, or am I missing something?

Individual tern tracks never exceeded 2 minutes, we have now added the mean and max duration of the tracks in the results section and also added more information in the supplementary material, please see Figure S3 and Table S1). Further, sample size (individual tracks) was not an issue (>800 tracks before filtering of trajectories with a minimum length), yet capturing a range of tidal flow velocities was, therefore 2-minute hovers deemed sufficient to track a number of terns in varying flow conditions.

Line 145: it might be useful to include a table with all hovers, and the corresponding times and number of tern tracks recorded.

We agree that such summary statistics would be helpful and have now added a comprehensive table in the Supplementary material (Table S1).

Line 148: what did the calibration involve?

The UAV camera was calibrated in the lab using a standard checkerboard method, this has now been added to Line 143.

Line 170: what's the accuracy of the altimeter, and does it matter for subsequent analyses?

The accuracy of the barometric altimeter is typically +/-1 m, or 1% of the hover altitude. This error propagates linearly into the scaling of the images and hence positions of tracked objects.

Line 172: how was this window size selected and does it affect the results?

Calculating the velocity as the first or centre difference of the tracked positions along each trajectory introduces jitter caused by position errors. Here, the apparent position of the tracked tern, measured as the centroid of the moving object extracted from the frame-to-frame differencing, varies according to its changing shape which is a function of the tern's three-dimensional orientation in flight and wing beat. We therefore follow the standard Particle Tracking Velocimetry method of Luthi et al (2005) using the low-pass filter moving cubic spline approach. This low-pass filtering operation of the position signal has a cut-off frequency of 2.73 Hz that represents a good balance between retaining detail within the trajectory and removing high-frequency wing-beat variation. The mean wingbeat frequencies for Common and Sandwich terns are in the range 3.1 to 3.7 Hz (Wakeling and Hodgson, 1992). Therefore, the 11-point, 2.73 Hz low-pass filter used in our study to smooth the 30 Hz data is justified and does thus not over-smooth the data. We have now added some clarification in the methods section.

As an aside, the variation in wing-beat frequency that can also be extracted from the tracking data may be used in future studies.

Lüthi, B., A. Tsinober, and W. Kinzelbach. 2005. Lagrangian measurement of vorticity dynamics in turbulent flow. *J. Fluid Mech.* 528:87–118.

Wakeling BYJM, Hodgson J. 1992 Optimisation of the Flight Speed of the Little, Common And Sandwich Tern. *J. Exp. Biol.* 169, 261–266.

Line 179-195: could you provide some references in support of this section? Particularly with regard to the corrections made and the standard functions used.

We have now provided the according reference for the PIV section.

Line 198: because the coupling of bird and flow data is critical, I think this 'three-dimensional interpolation in space and time' requires some additional details.

Wording has been added to help clarify this in Lines 196-198.

Line 208: I would spell out 'absolute(curl)' for clarity.

We agree and have now changed this to absolute(curl) and also log(tortuosity) throughout the manuscript, this is also in response to another reviewer's comment.

Lines 228-231: is it worth describing model selection here?

Yes. Please note, in lines 232-233 in the Methods and lines 257-259 in the Results, we have now included the delta AIC values associated with the models without either of the two covariates (absolute(curl) and divergence). We did not conduct a formal model selection with respect to the number of HMM states, as it is well-known that this will point to models with (biologically) unrealistically large numbers of states, where the “additional” states can help to improve the goodness of fit – especially with patterns as complex as the one we are dealing with – but can usually not be meaningfully interpreted (Pohle et al. 2017).

Pohle J, Langrock R, Beest FM Van, Schmidt NM. 2017 Selecting the Number of States in Hidden Markov Models: Pragmatic Solutions Illustrated Using Animal Movement. *J. Agric. Biol. Environ. Stat.* **22**, 270–293.

Lines 239-240: I think this clarification should be moved to the Methods, where the two putative states are first mentioned (line 217?).

As suggested, we have moved the interpretation of the two states to the methods section of the HMM (L216-218).

Lines 246-247: in general, I’m OK with the identification of those two discrete states. However, from Figure 2, it looks to me like there could be a more continuous change in speed and tortuosity, as reflected in the sample path shown and the not-really-bimodal state-dependent distributions. The option of fitting an alternative state space model with a continuous state may warrant a comment in the Discussion.

We agree that a distinction into two distinct discrete states is somewhat simplistic. The tortuosity time series (and also the turning angle time series now included in the supplementary material) do indicate two discrete states (effectively straight vs. curved flight path), however there are also rather smooth transitions between these states – which makes sense physically and biologically. Corresponding model extensions, either using a continuous state space (possibly nested within discrete states operating at a coarser scale), or autoregressive terms in the observation process, could potentially improve the fit with respect to these smooth transitions, but would be numerically less stable and would complicate the interpretation.

We added a detailed paragraph in the discussion (L 370-391) in which we discuss the simplifying model assumptions we make as well as possible alternative modelling approaches. Given the focus on the dynamics of the state process, we argue that our simple model is adequate as it produces biologically plausible states (even if not capturing all of the serial correlation). Subsampling at a lower temporal resolution, as suggested by Reviewer #2, is also an option for reducing the serial correlation in the movement metrics, and hence potentially to some extent get rid of the more continuous variation. We explore this option in the electronic supplementary material (Fig. S5).

Lines 251-254: this sentence is a bit convoluted, I suggest some rephrasing; also, I would avoid repeating what was described in the Methods.

We have re-written the sentence slightly. However, we think this sentence helps the reader to interpret the figures.

Line 272: again, I don't think you provide sufficient justification for why terns should be considered model surface foragers; I would expand on this point, or omit.

We agree and have re-written this sentence as follows, L285-286:

'Our drone-based approach, tracking seabirds and underlying physical features in synchrony, revealed new insights into localised tern foraging strategies amongst turbulence.'

Line 282: do you feel comfortable extending your results to all seabirds? I can think of species that simply sit on the surface and feed from there, or species that use olfaction and fly much closer to the surface than terns. I imagine these species will show a different interaction with physical cues.

We agree and have completely re-written the first two paragraphs of the Discussion.

Line 285: a bit cryptic – is it worth explaining what these processes are?

We believe that the references provided are sufficient to indicate that seabirds can turn into an active foraging mode simply by either stealing prey from other seabirds or by seeing conspecifics foraging.

Line 298: as per my comment above, I would be careful not to draw conclusions on all seabirds, which can move and hunt using very different strategies, from these results on terns studied in a peculiar context (and on a limited number of sampling occasions).

We agree and have changed 'seabirds' to 'terns'. However, we also like to add here that the context is not peculiar. Turbulent features like the ones under investigation here are very common in dynamic coastal environments along the breeding range of terns. And these areas are also those experiencing the highest degree of anthropogenic change, such as ocean energy installations, bridges and harbours which, in combination with faster flows, create regions of turbulence.

Line 306: what does 'its' refer to?

It refers to the animal's ecological function (e.g. nutrient recycling). We have now re-written this sentence to, L329-332: 'Ultimately, investigating how an animal's perceptual abilities determine how it extracts information from the environment, is an essential component of their foraging ability and thus, the animal's ecological function'.

Line 307: formulate or formalise?

Thank you, we have changed this to 'formulate'.

Lines 310-311: I don't understand this sentence, could you clarify?

We have changed the order and wording in this paragraph slightly to clarify that targets or cues may be perceived from a distance and the visual challenge being, that terns need to continually extract and process information from their environment during flight, L335:

' In-flight terns must continually extract and process information from their environment which includes the visual challenge of locating an environmental cue at some distance which may be indicative of prey items.'

Line 334: in the sense that tagged animals may not necessarily go through the area of interest? I would slightly rephrase to clarify.

Yes, we have clarified this, L359-360: ' For instance, animal-borne telemetry applied to a few individuals may not capture movement within a specific area of interest if they do not frequent the site.'

Lines 335-337: the syntax of this sentence needs some revising.

Thank you for pointing this out, we have now rewritten the sentence as follows, L360-363:

'Shore observations or vantage point surveys, may quantify the relative number of birds using an area, but the oblique angle of the observer hinders the matching of a bird's spatial position to a feature underneath.'

Lines 353-356: I fully agree, but I am not sure this is a good/relevant concluding statement for this work.

We agree and have removed this section.

Figure 4: I would include a legend for the colours in this (very nice!) plot.

Thank you, we have now moved this figure into the supplementary material and provide such details in the figure description.

Referee: 2

Comments to the Author(s)

Review: A bird's eye view on turbulence: Seabird foraging associations with evolving surface flow features

The authors present interesting research combining remarkably high detail tern tracking data and environmental data measured in situ using drones. Such analyses, if scaled to longer periods of time and more locations, could prove to be extremely valuable as the methods become increasingly accessible. Although the hypothesis and objectives were not clearly defined, the methods are adequate for presumed goal of the research; identifying effects of turbulent features of tern behaviour using drone-based telemetry. I first propose three moderate revisions followed by outlining smaller technical, grammatical, stylistic, and typographic revisions.

We appreciate the positive feedback and would like to thank the reviewer for the thoughtful comments and suggestions made in their report. We have addressed all comments made by the reviewer.

First, there needs to be a discussion and justify for the resolution that is used. The scale of the data determines the types of behaviours that can be identified. On one hand, coarse data may be prohibitive for studying finer-scale behaviours, however, at too fine a scale, different behaviours may begin to appear increasingly similar and be hard to differentiate. For example, at an extremely fine scale, the movement of nearly all behaviours appears perfectly straightened out. There are several factors that hint to me that the resolution may be too high for the desired behaviours.

At which resolution to model high-resolution raw data is indeed often a tricky question. Typical problems that arise at very high resolutions – pointed out also by the referee – are the potentially very high computational cost, statistical artefacts resulting from the high resolution (such as an inflation of turning angles exactly equal to zero), and generally more complex dependence structures (e.g. the conditional independence assumption within HMMs will typically be violated). However, given that any subsampling leads to a loss of information, we would generally argue that using the highest possible resolution is ideal *if* it is feasible to model at that resolution, i.e. if the problems just stated do not occur or are simply negligible/not relevant with respect to the study aim. In particular, any behaviour that can be inferred from a subsampled (low-resolution) time series can in principle also be inferred from the raw (high-resolution) time series – the information is clearly contained in the raw data, it's just that the modelling may be more challenging. Therefore, as none of the potential problems stated above occurred in our setting, at least not in a way that it would affect the inference on the state-switching dynamics (which is the focus of our analysis), we chose to model at the original 30 Hz resolution.

We do however agree with the referee that additional analyses at alternative (lower) resolutions, potentially also involving other movement metrics (e.g. turning angle instead of tortuosity) will strengthen the justification of our approach, and will be beneficial to readers interested in modelling ultra-high-resolution data. Following the suggestions of the referee, we have therefore included additional analyses in the supplementary material. More detailed discussion of the associated findings is given below in response to the specific comments and suggestions by the referee (as well as in the manuscript).

First, the use of tortuosity as a measure of turning (as opposed to turning angle), suggests that from point to point, movement is almost perfectly straightened out, and to detect state-specific differences in turning, it was necessary to calculate a turning metric across 11 locations (I also wonder whether this moving window for tortuosity violates the assumption of independent observation data in a basic HMM). Second, even with the use of tortuosity over turning angle, there is a very strong right skew of the tortuosity data in Fig. 2 B, which makes me suspect a lower resolution may exhibit a wider range of variation and make it easier to resolve distinct behaviours. Third, in Fig. 2 C there appear to be distinct flight paths at coarser scales may also be classified as “transit” and “active” foraging. It appears as though the current model is identifying individual turns from straightened section while there are sections of the path that are broad sweeping turns and sections of casting (i.e., “zig-zags”). Fourth, in Fig. 2 D it appears that there are many dozens of points within one bout of being in a behaviour, which may make it harder to identify the effects of covariates on the transition probabilities. This is supported by Fig 3 A and B, with the probabilities of remaining in the same state being exceptionally high (~0.99). The resolution that was used may indeed be valid for the behaviours investigated, however I suggest some more justification and discussion about the strengths and weaknesses of the resolution that was used. In addition, I would suggest including an appendix with the same analysis done on a lower resolution with corresponding versions of Fig. 2 and Fig 3, and a few sentences on similarities and differences. Such an appendix could be especially useful and insightful if future researchers are interested in investigating behaviours at slightly broader scales using similar ultra-high-resolution data, particularly if modelling directional bias or using step selection functions.

In the additional analyses conducted, we considered a lower resolution (2.73 Hz) and also turning angles instead of the tortuosity metric (in different combinations). The corresponding results are shown in the supplementary material (Fig. S5) to the paper. In summary, we find:

- that subsampling (to a 2.73 Hz resolution) slightly but not substantially changed the decoded state sequences, and did not change the overall results regarding the state-switching dynamics;

- that turning angles were difficult to model at the high resolution (30 Hz; the goodness-of-fit here was worse than for the other models, as effectively suspected by the referee), but OK to model at the low resolution (2.73 Hz), and that in both cases the corresponding model led to the same overall results regarding the state-switching dynamics as obtained when using tortuosity in the observed process.

Given that the setting is unsupervised – i.e. we do not have any training data containing information on the actual behavioural states – it is not possible to ultimately determine which of the approaches led to the “best” results (i.e. regarding the minor discrepancies in the state decoding, it is not clear which outcome here is “better”). We do in any case explicitly state in the paper that the HMM states are no more than *proxies* for active and transit foraging, respectively, a caveat that is in any case inevitable in such an analysis. However, the 30 Hz data does in our view include additional information with respect to behavioural variation: for example, instantaneous behaviours such as hovering, a quick scan to the left and the right, or other erratic behaviour, might be smoothed over when subsampling the data.

We do not see any problem with the fact that probabilities to remain in a state are close to 1. This is obviously a consequence of the high resolution: for any given time period, the probability of switching states will be virtually identical across resolutions, as with the higher resolution there are simply more occasions at which a switch may take place. Thus, the resolution at which the data are modelled does not affect the implied state-switching dynamics, except of course for the relatively minor differences in the estimates.

Regarding the violation of the conditional independence assumption made within the HMM: yes, this assumption is certainly violated, though not just because of the moving-window approach to calculating tortuosity – the two time series to be modelled, whether at 30 Hz or 2.73 Hz, show very high serial correlation, also within states, and this is something which is simply not accounted for in a basic HMM. It would in principle be possible to include autoregressive components in the observation process to potentially improve the fit in that respect. We did try this, but were not successful – the corresponding model formulation turned out to be immensely unstable numerically, and we were not able to find a model with a decent fit to the data. Conceptually even more appealing would in fact be a hierarchical HMM, with continuous states at the sub-second scale, embedded within discrete states operating at a slightly coarser scale

In any case, such extensions in our view are not necessary given the focus on the state-switching dynamics. In other words, as long as the model yields a plausible state classification, which we argue is clearly the case, it does not really matter if there is some lack of fit in the observation process. This would obviously be different if the focus was on predicting future values of the time series, as would typically be the case in financial applications.

We have added a paragraph in the discussion section (Lines 370–391) where we discuss these issues related to our somewhat simplified modelling approach, mentioning the alternatives and also explaining why we chose to stick with our model.

Second, I do not think that selecting the delays for the time-to-contact using likelihood necessarily tells us the scale at which the features affect behaviour. Instead, I think the likelihood approach for delay selection tells us simply the scale at which most variation in the data is explained. However, stimuli likely have unique effects on behaviour that are different at different scales (i.e., different delays), and the effects (or lack thereof) at one delay provide uniquely important information from effects at another delay. For example, abs(curl) may have an ecologically significant effect on behaviour at a short time-scale as at a longer scale, however simply because the statistical likelihood is different, does not mean one is invalid. Therefore, I think there is merit to discussing the effects of vorticity and divergence at multiple delays, not only where the combination of delays produces the maximum likelihood. Especially since there are some very different delays that have a similar likelihood, for example, at a delay for abs(curl) of 0.25, a divergence delay of $d=0.5$ has comparable delta likelihood (-4.99) to a divergence delay of $d=4$ (-2.94); statistically relatively as powerful, but ecologically quite different. Specifically, I think it may be worth while to very briefly highlight 3 or 4 combinations of delays that have a decent likelihood but are far apart in table S1. It seems strange that upwelling ahead of the track cues behaviour to stay in transit, but does not cue to localised foraging when arrive at the source of upwelling.

This is a very good point, and we agree with this interpretation of the delays. The likelihood comparison merely identifies the scales at which most variation in the data is explained, and we have now carefully revised the text in the results and discussion accordingly. Rather than simply using an arbitrary delay (and therefore, distance) to a feature, we chose to use this approach to help interpret where most of the variation lays. As outlined in the discussion of our manuscript, it is not known how seabirds perceive dynamic cues during flight. For instance, there is some discussion that seabirds also scan ahead of their flightpath and therefore, simply extracting variables underneath the terns' positions would have been too simplistic.

In line with the reviewer's comments, we have modified the second sentence opening the discussion, L288-290:

'As predicted, tern movement patterns showed a strong associations with specific evolving turbulence features and these varied with the time-to-contact (as expressed in delays), indicating the scale at which most variation in the data was explained.

We have also added this to the results, in L263-271: 'It was not known a priori how terns would perceive dynamic cues during flight, and these values identified the scales at which the variation in the data, and specifically the probabilistic switching between the two states was best explained. Vorticity extracted almost underneath the terns and divergence ahead of the flight path thus yielded the model with the best goodness-of-fit. This does not necessarily imply that terns primarily respond to features at these time-to-contact values, and several other delay combinations yielded maximum log-likelihood values not much smaller than the optimum at $d=0.25/2.0$ (maximum log-likelihood valus

were substantially lower when using higher delays d for absolute(curl), but not much lower for any d from 0-5 for divergence).'

Finally, from an ecological perspective, it makes sense for terns to perceive divergence more at a distance, cueing transit behaviour, more so than vorticity magnitude, as upwellings are much more conspicuous, smoothing a rough surface. A newly erupting boil will be highly conspicuous. However, they do evolve at second-scales, therefore on approach, this boil will have already changed in scale in intensity. Please see our additional explanations in lines 296-305 in the Discussion.

Last, there are four smaller components that I think need to be more developed. First, the hypothesis and prediction should be stated more explicitly in the introduction and the discussion should link to these specifically. Presently, there is no hypothesis in the introduction, but a prediction that takes the form of a hypothesis and no true prediction. Furthermore, the discussion references a hypothesis not stated in the introduction and one that is not precisely linked to the prediction in the introduction.

We agree and thank the reviewer for pointing this out. We have now refined the hypothesis and prediction (please see details comments below) and link back to these in the discussion.

Second, the abstract and the discussion in general could benefit from more interpretation of the results, and the impact of findings to our ecological understanding of terns. Particularly because this research is framed in predicting responses to environmental change and unlocking knowledge gaps in seabird sensory foraging ecology. The interpretation of the results does not seem adequately tied back to these objectives. Does this add any new specific insights about this specific species or coastal feature? There are elements of this throughout the discussion but not a succinct focal point.

We have now added a more in-depth discussion and interpretation highlighting the bio-physical importance of our results, please refer to our more detailed responses below. As a quick summary, it shows that the occurrence of conspicuous boils triggers the investigation of such features (transit mode) and terns were likely to start foraging over vorticity. Both these turbulence features are not only common around natural wake features (e.g. flow going past an island), but occur when tidal flows go past monopiles (e.g. wind farms or the tidal energy monopile in our study):

1. Grashorn, S. & Stanev, E. V. Kármán vortex and turbulent wake generation by wind park piles. *Ocean Dyn.* 66, 1543–1557 (2016).
2. Ouro, P., Runge, S., Luo, Q. & Stoesser, T. Three-dimensionality of the wake recovery behind a vertical axis turbine. *Renew. Energy* 133, 1066–1077 (2019).

We thus indicate towards the end of the Discussion (L393-397) that environmental change, such as the introduction of anthropogenic structures, has the potential to affect foraging opportunities, as structures can influence the occurrence, scale and intensity of turbulence features.

Third, there is currently no discussion about limitations of the methods or results, and as a result, no specific suggestions for future research (either ecological or methodological). For example, I would have liked to see some discussion of other methods that might provide complimentary evidence of turbulent feature selection. For example, HMMs with directional bias toward targets, or using step selection analyses to explicitly examine selection, which HMMs cannot do as they do not consider available but unused habitat.

We have now added discussion points regarding the limitations of the methods applied and have also added more specific suggestions for future research. Thank you for these suggestions. Please see the new paragraph in lines 370-392.

Last, I think the authors should use the “state probability” or “stationary state probability” rather than “state occupancy”. “Occupancy” has a very specific definition with regard to animal movement/location data (see Lele et al. 2013. Selection, use, choice and occupancy: Clarifying concepts in resource selection studies.). Instead, I advise using “State probability” or “stationary state probability” (as the authors used in Fig. 3 caption), which would be more consistent with the terminology used in the HMM literature.

Good point, we agree and hence have changed the wording to “stationary state probability” throughout the paper.

Specific feedback:

L24. Can remove “marine predator” for breadth or add “oceans” as on L43.

We have changed ‘marine predator’ to ‘seabird’.

L25. Not self-evident that “turbulence” is referring to that of water and not air (especially since the title indicates the focal species is avian)

We appreciate the possible confusion (fluid versus air motion) and have clarified this as follows: ‘For instance, turbulence in the water arising from natural or anthropogenic structures can affect foraging opportunities in tidal seas.’

L26. Example of “turbulent features”?

We have added an example of turbulent features: ‘ Yet, identifying ecologically important localised turbulence features (e.g. upwellings ~10-100 m) is limited by observational scale and this knowledge gap is magnified in volatile predators.’

L28. “vulnerable to coastal change” feels out of place.

This was in relation to environmental change in the first sentence, but we have removed this (please see correction above).

L29. Specify the exact number of trajectories.

We have now specified the exact number of trajectories (n=143).

L30. “earliest evidence for predator associations with localised physical foraging cues” is extremely vague, and is certainly untrue. There is a copious amount of research on “predator associations with localised physical foraging cues”

We agree and acknowledge that this is too vague. We have changed this to: ‘We thereby provide the earliest evidence that localised turbulence features can provide physical foraging cues’. We’d like to clarify here also that predator associations with turbulent cues have been shown (e.g. with meso-scale eddies), however not on the localised scales under investigation here. Please note that we have also specified the scales of the features as per the response above (L26).

L32. Indicate that this was based on several* species of terns? And consider noting the general region.

In addition to specifying the exact numbers of trajectories (as per comment above), we have now clarified that our study was representative of several tern species and have reworded the sentence as follows: ‘ Here, using a drone-based approach, we present the tracking of surface-foraging terns (143 trajectories belonging to three tern species) and dynamic turbulent surface flow features in synchrony.’

L34. “transit foraging” is undefined and not a colloquial behaviour.

To make the abstract more comprehensive to a broader audience, we have changed this to ‘transit behaviour’. However, our study describes foraging/movement strategies within a foraging aggregation (terns deemed to be foraging), therefore, we are moving beyond the conventional definition of two-state models (“foraging” and “transit”). This is why earlier in the abstract, we explicitly outline that we tracked ‘surface-foraging’ terns, as traditional ‘transits’ (straight-line paths) were removed for this analysis and only terns deemed to be foraging were included.

L37. “understand [the effect of] coastal change [on...]”. The research does more than just understand coastal change.

We have changed the wording of this final sentence of the abstract to: ‘Finally, it lays the foundation to predict responses to coastal change to inform sustainable ocean development.’

L44-48. Consider switching second and third sentences

Thank you for this suggestion, but we think there is a natural progression in the order of the sentences: environmental change (which can be either, climate change or coastal development) -> anthropogenic activities -> installations supporting the blue economy.

L48. "This" is vague

'This' refers to coastal change. We have added wording in the text.

L49. Do such changes only generate new foraging opportunities? Or can they threaten them as well?

We have now rewritten this sentence to clarify that interactions are likely, but that we do not know how they may affect foraging success, an important parameter to understand population change: 'This coastal change is undoubtedly leading to new interactions between marine predators and installations and we are yet to understand how this may influence foraging success (Scott et al), with some evidence that installations can even generate new foraging opportunities [original references].'

Please also note the insertion of a new references (Scott et al):

Scott BE, Langton R, Philpott E, Waggitt JJ, Langton R, Waggitt JJ. 2014 Seabirds and marine renewables: are we asking the right questions? In Marine Renewable Energy Technology and Environmental Interactions (eds MA Shields, ALL Payne), pp. 81–92. Orkney, Scotland: Springer Netherlands. (doi:10.1007/978-94-017-8002-5_7)

L50. Last sentence feels long clunky. Streamline or split into two?

We agree and have now split this into two sentences by changing the order as follows: 'Foraging strategies may vary in response to physical changes in local conditions. Assessing how free-ranging animals adjust and fine-tune their foraging movements in highly complex and dynamic environments is therefore fundamental to understanding how they may respond to anthropogenic change.'

L61. Incorrect dash used

We have corrected the dash.

L68-69. Not clear why the examples are numbered. There are likely other physical processes that emerge in tidal environments, and you do not refer back to these two features explicitly.

That's correct and we have changed the sentence accordingly: 'Here, strong currents interacting with fine-scale heterogeneity in bathymetric features or man-made structures can give rise to numerous physical processes; including localised features (e.g. boils [21] (localised upwellings), convergences, eddy vortices), and dynamic boundary waters (e.g. shear lines and flow reversals).'

L70-73. "some of these turbulent features..." This sentence describes how turbulent features specifically increase prey, which is first mentioned in the second sentence. could this sentence be moved up to the 3rd sentence?

We have changed the order of the sentences accordingly.

L78. Specify the scale implied by “highly localised”.

We have now specified the scale in accordance with the first mention of ‘localised scales’: ‘(~10-100 m)’.

L78. Why the qualifier of the prey having to be “conspicuous”?

We agree and have removed ‘conspicuous’.

L82. Could “mechanisms underlying foraging strategies” be made more specific. I appreciate the specificity in the previous sentence, which concretely highlights “physical properties” to which seabirds “show affinity to”.

We appreciate that *mechanisms* is too vague here, and have changed ‘mechanisms’ to ‘physical cues’ to be more specific.

L84. “such as [satellite-derived] animal telemetry [or animal tracking]”. The drone-based approach in this research is technically “animal telemetry” (remote measure/monitoring of animal). Could also reword this part of the sentence to “such as associating coarse satellite-derived data with higher resolution animal tracking.” This clarifies that the most common limitation is not necessarily the resolution of the track, but of associated environmental data, which is what the following sentence seems to be suggesting.

We agree with the suggestion and have re-formulated the sentence as follows, L84-87: ‘Yet, the required high spatio-temporal resolution (meters and seconds) to capture such associations is often unattainable using traditional approaches, such as associating coarse-scale satellite-derived data with higher resolution animal telemetry.’

L84. “Animal telemetry” should not be hyphenated.

We have removed the hyphenation.

L86-86. It is the spatiotemporal mismatch that leads to not capturing highly localised associations. Switch those two clauses. That is, something to the effect of “... averaged oceanographic data [results in a spatiotemporal mismatch...], and may obscure highly localised associations”

As suggested, we have changed the sentence structure as follows, L87-90: ‘ For instance, with the rapid dynamics associated with seabird flight, temporally or spatially averaged oceanographic data lead to a spatio-temporal mismatch between movement metrics and habitat characteristics at the visited cell and may thus not capture highly localised associations.’

L93. Join with previous paragraph if starting with “specifically”

The two paragraphs have now been joined.

L94. Can remove “during foraging” as this can shed light on any number of behaviours

This has been removed.

L96-97. A bird’s eye view on underlying physical features does not quantify context-specific behaviours. It identifies features that can be subsequently used to identify context-specific behaviours. Reword.

We have re-formulated this sentence to, L98-99: ‘Such visualisations would allow a bird’s eye view on underlying physical features, thereby aiding the quantification of context-specific behaviours [34].’

L97. Remove space after thereby.

Done.

L101-104. Could you reframe this as a formal hypothesis, then add a specific prediction at the end of L113? E.g., Change L101 to “We [hypothesise] that surface-foraging terns (*Sternidae*) vary their foraging... which serve as physical cues [of high prey density] during foraging”. The prediction should be the specific expected results of your analyses assuming your hypothesis is true. In this case, the prediction would be that the underlying turbulent features affect the state transition probabilities.

We agree with the suggested changes and have re-formulated our hypothesis as follows: ‘We hypothesised that surface-foraging terns (*Sternidae*) vary their foraging movement in response to localised coherent surface flow features, predominantly vorticity (the curl of the surface flow) and upwellings (regions of positive divergence), which could serve as physical foraging cues.’ Please also note that we have moved our prediction to L112, as suggested: ‘We predicted that state transition probabilities would be affected by the strength of the underlying turbulent feature as well as its distance, as perceived by the terns.’

L103. “Physical cues of___”

Please note, as per above, we have changed this to ‘physical foraging cues’.

L114-116. As noted earlier in the introduction, other research has tied foraging to turbulent features at broader scales, presumably “specific” at their scale (e.g., reference 5). Reword to indicate that this is the earliest to do so at the sub-10-100m scale, or that this is the finest scale yet for remote-tracking-based data.

We agree this was too vague and have re-formulated the last paragraph of the introduction as follows, L113-115: ‘This allowed us to quantify the influence of prevalent oceanographic features on a surface-foraging marine predator on hitherto unobtainable scales (~10 –100 m).’

L120. This is not a sentence that should be referenced.

Please note, we have now removed the entire sentence as requested by the other reviewer given that the format of the journal does not require an indication in the introduction of what will be discussed.

L148. What did the in-lab camera calibration entail? Consider removing if it was minimal.

The UAV camera was calibrated in the lab using a standard checkerboard method – it is an important component of the methodology and so has been retained.

L167. You use tortuosity here, but only define it on L174. The procedure should be defined for the first instance of its use.

Thank you for noting this, we have now moved the definition of tortuosity above when it is first mentioned.

L202-206. There is no need to introduce the “X” parameter, which is simply values of $d > 0$. It would be more concise to simply say “To investigate such ‘time-to-contact’ effects, time-offsets (delay $d \in \{0, 0.25, 0.5, \dots, 5\}$ in seconds) were applied, where $d = 0$ represents vorticity/divergence values extracted directly underneath the tern’s position and $d > 0$ represent values ahead along the flight path”

Thank you, we have made the corrections as suggested.

L204. “ $X = 0.25 - 5 \text{ s}$ ” is misleading and suggests $X = -4.75$

We have removed this, as per comment above.

L205-206. “To ensure parity between time-offsets, all tracks were truncated by the maximum offset of $d = 5 \text{ s}$.”

We have made the change as suggested: ‘To ensure parity between time-offsets, all tracks were truncated by the maximum offset of $d=5$.’

L212. Add space after ‘travelling’

Done.

L216. Why did the authors model $\log(\text{tortuosity})$ and not simply tortuosity-1? Presumably both of these would yield values > 0 .

In all honesty, $\log(\text{tortuosity})$ was simply what first came to mind. In any case, it does not make any difference, as the tortuosity values are all only slightly larger than 1, such that (2nd order Taylor expansion):

$$\log(x) \approx \log(1) + (x - 1) - \frac{1}{2}(x - 1)^2 \approx x - 1.$$

We did nevertheless re-analyse the data also using tortuosity-1, and the results were virtually identical.

L216. Why do you use “abs(curl)” and “(log-) tortuosity” and not “log(tortuosity)”? I find “(log-) tortuosity” confusing, it could be interpreted as log(-tortuosity) (which produces complex numbers in this case), -log(tortuosity), or log(tortuosity). Alternatively, If you exclusively use abs(curl) and log(tortuosity), you could clearly state this once and simply use “curl” and “tortuosity” thereafter.

We agree and now use absolute(curl) and log(tortuosity) throughout the paper.

L228. “d= 0-5.0” suggests d = -5. Either remove (as this has already been defined), or replace with “(i.e., delay $d \in \{0, 0.25, 0.5, \dots, 5\}$ s”

That’s correct and we have replaced this with ‘(d)’ as d was previously defined.

L228. Cite that you used the moveHMM package

We did not use moveHMM – everything was implemented from scratch, as many different model formulations were tried out, not of all which could be fitted using moveHMM. (Arguably, we could probably have used momentuHMM.)

L244. What is “opportunistically” referring to in the context of searching?

This refers to a behaviour of occasionally searching while in transit, describing a movement behaviour ‘as though opportunistically searching while in transit’ (in accordance with the Wilson et al. reference). For instance, terns may simply ‘sweep’ across an area because it is used by conspecifics for foraging, presenting the ‘opportunistic’ element. This transit search/foraging is still to be differentiated from direct flight (transit search=generally slower than direct flight and more directional changes, while not erratic). The direct flight is described as ‘a clear and consistent direction, usually fast, often adopted when flying back to the colony with a fish’. Such tracks were not considered in our analysis, as the aim was to analyse only terns that were deemed to be foraging or actively as well as opportunistically searching for food. Therefore, straight-line transits (direct flights) were omitted from the data set as described in the methods.

L246. Remove “clearly identifiable” as it is not objectively quantifiable

We have changed this to ‘distinct’.

L247. Space after “Fig.”

Done.

L251-252. Active foraging and transit foraging were previously defined as “state 1” and “state 2”, which therefore should be the notation used (i.e., “Pr(state 1 -> state 2)”, and “Pr(state 1)”). Or redefine states as TF and AF (transit foraging and active foraging, respectively).

We have clarified this using: ‘The state transition probabilities($\Pr(i \rightarrow j)$), for $i, j = 1, 2$), and as a consequence also the stationary state probabilities ($\Pr(i)$, for $i = 1, 2$) were modelled as...’

L252. Here and throughout, I would advise against the use of “state occupancy” and instead use “state probability”. “Occupancy” has a very specific definition in the field of movement ecology (see Lele et al. 2013. Selection, use, choice and occupancy: Clarifying concepts in resource selection studies.). “State probability”, “stationary state distribution”, or “stationary state probability” (as you have in Fig. 3 caption) would be more consistent with the terminology used in the HMM literature.

The wording has been changed throughout to “stationary state probability”.

L258. would $Pr(i \rightarrow j)$ be more consistent with previous notation? ‘X’ was previously defined as $d > 0$.

Yes, and we have removed the X definition as suggested in the reviewer’s other comment, so this does not require a change.

L265. Consider: “Stationary state probability [Pr(i)]”

This has been changed according to the suggestion.

L274. “; d in seconds” is redundant with previous definitions

We agree, and have removed this as suggested.

L279: “... [upwelling regions] do not necessarily evoke foraging activity”: this has not been demonstrated in this research. See general comments about presenting results of $d = 0.25$ for upwelling.

We have removed this. We have re-formulated this statement according to our results, L296-298: ‘Therefore, conspicuous upwellings may provide a strong physical cue even at some distance, leading to the investigation of such features.’

L283. More specific on which local scales – “such localised scales” is ambiguous.

We have removed ‘such’ and with previous specifications on the scale definitions, this should now be more clear.

L289. This hypothesis was not previously articulated and is different from the prediction on line 101.

We have now adjusted this in accordance with the hypothesis, L286-288: ‘We hypothesized that terns may vary their foraging movement in response to localised coherent surface flow features, which could serve as physical foraging cues.’

L308. Remove space after impacts.

Done.

L314. Replace “=” with “i.e.,”

Done.

L320. Do not start a paragraph with “therefore”. Join with previous paragraph.

We have joined the two paragraphs as suggested.

L343. Replace “e.g.” with “, for example,”

We have removed ‘e.g.’.

L348. “while simultaneously [estimating] ecological...”

We think ‘providing’ is appropriate here as we have concurrently mapped the entire flow field, however we have changed ‘physical environment’ to ‘flow field’.

L554. Fig 2 C A starting location would be interesting to denote (particularly to compare with the time-series).

We have now indicated the starting position using a star symbol.

L599. 1 Fig 4 is not referenced in text and can be moved to the appendix.

We agree and moved this to the electronic supplementary material. We make reference to the figure in L310.

L599. 2. Denote start location along track.

We have addressed this in the supplementary material.

L599. 3 This is an interesting chart, but it is difficult to identify much meaningful patterns. The authors could indicate in the caption the main pattern that is being taken from the figure (e.g., transit toward vortex, followed by a transition to active foraging when surface feature is reached). It is quite hard to identify the height of surface features, particularly in mass on the left. Perhaps make the features much more translucent, but with outlines (with transparency approximately the same as the current areas).

We have amended the figure legend accordingly.

Appendix B

Response to Referees Letter , followed by manuscript showing the tracked changes

Associate Editor Board Member

Comments to Author:

Dear authors,

One of the original reviewers and myself have read your revision, and we are both happy with the rigorous way how you have dealt with the previous comments. Some relatively minor points remain about the accessibility of code and data and some grammatical points raised by the reviewer.

Dear Professor Hans Heesterbeek,

We are delighted to see the review of our manuscript and have now revised the manuscript accordingly. We'd like to clarify that our initial submission contained the data and code and we have since moved this to Dryad: https://datadryad.org/stash/share/7c_AVu8r4EbmR9eco09S3vPuVS8FJNONMISrrtSln5M Please accept our apologies as I have not uploaded these files during the last re-submission. We have made both, the data and the code, available to reproduce the study's results (analysis and figures).

Reviewer(s)' Comments to Author:

Referee: 2

Comments to the Author(s).

General Comments:

I thank the authors for taking time to address all of the points raised in my first review and I believe the revised manuscript is significantly improved! I am satisfied with the response to my review, and I particularly appreciate the additional methods investigating a lower resolution and the use of turning angle. Although it did not change the results, it eliminates a potentially significant point of uncertainty and provides a starting point for future analyses. If data and R code is provided as supplementary material (which I do not currently see), you should note this somewhere in the manuscript. I support publication of the manuscript following minor grammatical revisions listed below.

We'd like to thank the reviewer for the in-depth review and extremely helpful and constructive comments. We agree that the revised and final version has greatly improved. As suggested by the reviewer, providing the additional outputs at varying resolutions and movement metrics reduces uncertainty and strengthens our approach and the results of the study. The data and code has been deposited to Dryad and will thus be made available upon publication:

https://datadryad.org/stash/share/7c_AVu8r4EbmR9eco09S3vPuVS8FJNONMISrrtSln5M

Specific comments:

LL48-51. Sentence is a bit of a run on. Perhaps new sentence after “predators and installations” and maybe start with “There is some evidence that installations can..., however we are yet to understand how...”

We have modified the sentence structure as follows: “This coastal change is undoubtedly leading to new interactions between marine predators and installations. While we are yet to understand how this may influence foraging success [4], there is some evidence that installations can even generate new foraging opportunities [5,6]. “

LL66. Remove comma before “as a result of”

We have removed the comma as suggested.

LL66. Consider moving the citations to the end of sentence.

Thank you, we have moved the references to the end of the sentence and inserted the word 'possibly': "Such turbulence can provide physical mechanisms to enhance prey accessibility, possibly as a result of prey displacement in the water column, through turbulent vertical transport or physical aggregation at the surface (e.g. at the edges of features) [6,19-21]."

LL67. Add a comma before "or physical aggregation..." (as on LL 92 and elsewhere in the paper)

Thank you, we have inserted the comma here and below, as suggested.

LL73. Remove comma before "and dynamic boundary..."

The comma has been removed.

LL98. Add quotes (I think single) around "bird's eye view"

We have inserted single quotes.

LL103/112. Much improved hypothesis/prediction. Thank you.

Glad to hear- and thank you, this has much improved the clarity of the paper.

LL112. Remove ", respectively"

We have removed 'respectively'.

LL270. "d=0.25/2.0" suggests d = 0.125. Either clarify with something along the lines of "d=0.25 for curl and d=0.5 for divergence" or remove all together as it was mentioned three sentences prior.

Thank you, we have removed this '(d=0.25/2.0)' altogether, as yes, it has been mentioned in the sentence before.

LL270-272. Make the brackets into a complete sentence and add a reference to the supplementary material table showing this.

As suggested, we have changed this to: "Maximum log-likelihood values were in fact substantially lower when using higher delays d for absolute(curl), but not much lower for any d from 0-5 for divergence (Table S2, electronic supplementary material).

LL287-291. Good H/P.

Thank you!

LL326. Did you mean "sensory ecology" rather than "sensor ecology"?

Well spotted, thank you. We have corrected this to "sensory ecology".

LL328. Can remove brackets around "through physical accumulation".

We have removed the brackets, as suggested.

LL371-392. Excellent addition to discussion and gives lots to think about!

Thank you! We agree and are grateful for all the in-depths comments from the reviewer during the previous revision.